# Effect of prescribed sea-surface conditions on the modern and future Antarctic surface climate simulated by the ARPEGE AGCM

Julien Beaumet[1], Michel Déqué[2], Gerhard Krinner[1], Cécile Agosta[3,4,1], and Antoinette Alias[2]

[1]Univ. Grenoble Alpes, CNRS, Institut des Géosciences de l'Environnement, F-38000, Grenoble, France
[2]CNRM, Université de Toulouse, Météo-France, CNRS, Toulouse, France
[3]F.R.S.-FNRS, Laboratory of Climatology, Department of Geography, University of Liège, B-4000 Liège, Belgium
[4]Laboratoire des Sciences du Climat et de l'Environnement (IPSL/CEA-CNRS-UVSQ UMR 8216), CEA Saclay, F-91190 Gif-sur-Yvette, France

*Correspondence to:* Julien Beaumet (Julien.Beaumet@univ-grenoble-alpes.fr)

**Abstract.** Owing to increases in snowfall due to higher saturation water vapor pressure, the Antarctic Ice Sheet surface mass balance is expected to increase by the end of current century. Assuming no associated response of ice dynamics, this will be a negative contribution to sea-level rise, potentially in part compensated for by increased meltwater runoff. However, the assessment of these changes using dynamical downscaling of coupled climate models projections still bears considerable

uncertainties due to poorly represented Southern Ocean surface conditions and southern high-latitude atmospheric circulation. This study evaluates the Antarctic surface climate simulated by a global high-resolution atmospheric model, and assesses the effects on the simulated Antarctic surface climate of two different sea surface condition (SSC, i.e. sea surface temperature and sea-ice concentration) data sets obtained from projections with coupled climate models. The two coupled models from which SSC are taken, MIROC-ESM and NorESM1-M, simulate future Antarctic sea ice trends that are at the opposite ends

of the CMIP5 RCP8.5 projections range. The atmospheric model ARPEGE is used with a stretched grid in order to achieve an average horizontal resolution of 35 kilometers over Antarctica. Over the historical period (1981-2010), ARPEGE is driven by the historical SSC from MIROC-ESM, NorESM1-M CMIP5 historical runs, and by observed SSC. These three simulations are evaluated against the ERA-Interim reanalyses for atmospheric general circulation, the MAR regional climate model, and *in-situ* observations for surface climate.

For the 2071-2100 period, SSC from the same coupled climate models forced by the RCP8.5 emission scenario are used both directly and bias-corrected with an anomaly method. We evaluate the effects of driving the atmospheric model by the different choice of SSC from coupled models as well as the effects of the method (direct output and anomalies) used. For the simulation using SSC from NorESM1-M, no significantly different climate change signals over Antarctica as a whole are found when bias-corrected SSC are used. For the simulation driven by MIROC-ESM SSC, an additional increase of +170 Gt.yr$^{-1}$ in

precipitation and of +0.8 K in winter temperatures for the Antarctic Ice Sheet is obtained with bias-corrected SSC.
Antarctic warming and precipitation increase obtained in this study fall within the range of the CMIP5 ensemble RCP8.5 projections. For the range of Antarctic warming found (+3 to +4 K), we confirm that snowfall increase will outweigh expected increases in melt and rainfall. Using the end members of sea ice trends from the CMIP5 RCP8.5 ensemble projection, the difference in warming obtained ($\sim$ 1 K) is clearly smaller than the spread of the CMIP5 Antarctic climate projections. This

confirms that the errors in the representation of the South Hemisphere general circulation in the atmospheric models are also determinant for the diversity of their projected late 21$^{st}$ century Antarctic climate change.

*Copyright statement.* TEXT

## 1 Introduction

Projected 21$^{st}$ century increases of the Antarctic surface mass balance (SMB), due to higher precipitation rates, are expected to partly compensate for eustatic sea level rise (SLR) due to opposite changes in almost all other components affecting global sea level (Agosta et al., 2013; Ligtenberg et al., 2013; Lenaerts et al., 2016). However, the acceleration of ice flow and the interactions between oceans and ice shelves are expected to lead to an overall positive Antarctic contribution to SLR (Pollard et al., 2015; Ritz et al., 2015). Uncertainties in ice dynamics and surface mass balance trends are large and influence each other (e.g., Winkelmann et al., 2012; Barrand et al., 2013). It is therefore crucial to produce high-quality Antarctic climate projections for the end of the current century with reduced uncertainties, yielding trustworthy estimates of the contribution of the Antarctic Ice Sheet (AIS) SMB and useful driving data for ice dynamics and ocean-ice shelf interaction model studies.

Detection of an anthropogenic climate change signal is more challenging than in the Arctic. While some parts of West Antarctica and of the Antarctic Peninsula (AP) have experienced one of the world's most dramatic warming in the second part of the 20$^{th}$ century (Vaughan et al., 2003; Bromwich et al., 2013), there was no significant recorded temperatures trend in East Antarctica as a whole (Nicolas and Bromwich, 2014) except for some coastal regions that experienced a cooling in autumn over the 1979-2014 period (Clem et al., 2018). Moreover, the observed strong warming trend in the AP has shown a pause or even a reversal for 13 years in the beginning of the 21$^{st}$ century (Turner et al., 2016). Contrary to the dramatic sea ice loss observed in the Arctic (e.g., Stroeve et al., 2012), significant positive trends have been observed in the Antarctic sea ice extent (SIE) since the 1970s (Comiso and Nishio, 2008; Turner et al., 2015, e.g.), although recently record sea ice loss was observed in 2016/7 (Turner et al., 2017). Most of the Coupled Atmosphere-Ocean Global Circulation Models (AOGCM or CGCM), such as those participating the Coupled Model Intercomparison Project, Phase 5 (CMIP5, Taylor et al. (2012)) struggle to reproduce the seasonal cycle of SIE around Antarctica, and very few of them were able to reproduce the positive trend observed in the end of the 20$^{th}$ century (Turner et al., 2013). This is problematic because Krinner et al. (2014) showed that atmospheric model simulations of the Antarctic climate are very sensitive to the prescribed sea surface conditions (SSC), that is, sea surface temperatures (SST) and sea-ice concentration (SIC). Additionally, the amount of sea ice present in historical AOGCM climate simulations is strongly correlated to the absolute sea ice decrease in the projections for the 21$^{st}$ century (Agosta et al., 2015; Bracegirdle et al., 2015). This itself is strongly linked to the strengthening of the westerly wind maximum (Bracegirdle et al., 2018).

It is expected that the signal due to the current anthropogenic climate change will take over the natural variability of Antarctic

climate by the middle of the twenty-first century (Previdi and Polvani, 2016). Favier et al. (2017) and Lenaerts et al. (2019) provide more complete reviews of the current understanding of the regional climate and surface mass balance of Antarctica and of the key-processes that determine their evolution.

The dynamical downscaling of climate projections such as those provided by coupled models from the CMIP5 ensemble is generally produced using Regional Climate Models (RCM). The marginal importance of atmospheric deep convection for Antarctic precipitation does not require dynamical downscaling at very high resolutions. Therefore the use of a cloud resolving atmospheric model configurations is not necessarily particularly relevant for Antarctic climate projections. However, the added value of higher horizontal resolutions, such as the CORDEX-like simulations (Giorgi and Gutowski, 2016) at 0.44°, with respect to driving climate projections at coarser resolution (1 to 2°) from the CMIP5 ensemble is significant in coastal regions near the ice-sheet margins or on the AP, as the steep topography induces a strong precipitation gradient between wet coastal regions and dry inland East Antarctic Plateau (EAP). Below 1000 m above sea level (a.s.l), the origin of precipitation on the AIS is mostly orographic (e.g., Orr et al., 2008). For present-day climate, Lenaerts et al. (2016, 2018) found no significant differences in area integrated SMB and coastal-inland snowfall gradient between simulations with the RACMO model run at 5.5 and 27 km horizontal resolution. Genthon et al. (2009) similarly found reduced impact of the model grid resolution when excluding very coarse ($> 4°$) model of the CMIP3 ensemble. For future climate projections however, much larger precipitation increases were reported when using climate models at higher horizontal resolutions (Genthon et al., 2009; Agosta et al., 2013). The modelling of strong katabatic wind flows blowing at the ice sheet surface is also generally improved with a better representation of the topography (e.g., van Lipzig et al., 2004).

In this study, we use CNRM-ARPEGE, the atmosphere general circulation model (AGCM) from Météo-France, with a stretched grid allowing an average horizontal resolution of 35 km over the Antarctic continent to dynamically downscale multiple coupled climate simulations. As a global atmospheric model, ARPEGE is driven by prescribed SSC, but does not require any lateral boundary conditions. More details on the ARPEGE model setup are given in section 2.2. This method has some advantages over the more commonly used limited-area RCM method which depends, for future climate projection, on the quality of the representation of the climate of the region of interest by the driving GCM used at lateral boundary. When using stretched grid AGCMs, it is possible to use observed SSC at the present and model-generated SSC anomalies for projections (e.g., Krinner et al., 2008). When such an anomaly method is used, it is not absolutely required that the AOGCM used as a driver for SSC "perfectly" represents the atmospheric general circulation and its variability in the region of interest. Using a stretched grid GCM also allows us to better take into account potential feedback and teleconnections between the high-resolution region we are interested in, and other regions of the world. Several studies showed that AGCMs produce a better representation of atmospheric general circulation and a better spatial distribution of precipitation when forced by observed, instead of simulated SSC (Krinner et al., 2008; Ashfaq et al., 2011; Hernández-Díaz et al., 2017). Consistently, these studies also showed that AGCM runs for future climate with bias-corrected SSC yielded significantly different results than runs with SSC directly taken from coupled model output.

In this work, the bias-correction of SSC using a quantile mapping method for SST and an analog method for SIC is achieved following the methods and recommendations described in Beaumet et al. (2019). We drive the ARPEGE AGCM (Déqué et al., 1994) with both observed and simulated (from coupled models) SSC for the recent past (1981-2010). For future climate projections (late 21st century), we drive the model with SSC directly taken from two coupled models and with corresponding bias-corrected SSC. One aim of this paper is to evaluate the capability of ARPEGE at high resolution to represent the current Antarctic climate. Additionally, we quantify the sensitivity of present and future simulations with this AGCM to the prescribed SSC. The results are compared to those of similar previous studies. This study also differs from Krinner et al. (2008, 2014) as the ARPEGE AGCM is run at a substantially higher horizontal resolution (35 km) than the LMDZ model, which was used in these previous studies aiming at analyzing the impact of prescribed SSC on the Antarctic climate simulated by AGCMs.

Section 2 presents a short analysis of CMIP5 SST and SIE in the Antarctic region that was used as a basis to select the coupled model providing the SSC used here. This section also presents the ARPEGE model set-up used in this study. In section 3.1, we assess the ability and limitations of CNRM-ARPEGE to represent current Antarctic climate. Results and comparisons for Antarctic future climate projections are shown in section 3.2.

## 2 Data and Methods

### 2.1 Sea Surface Conditions in CMIP5 AOGCMs

Sea surface conditions have been identified as key drivers for the evolution of the climate of the Antarctic continent (Krinner et al., 2014; Agosta et al., 2015). In this study, SSC obtained from CMIP5 projections are bias-corrected using recommendations and methods from Beaumet et al. (2019) before being used as surface boundary conditions for the atmospheric model. Therefore, the importance of the realism of each CMIP5 model for the reconstruction of oceanic conditions around Antarctica in their historical simulation is reduced. There is however a limitation in the previous statement, as the analog method used to bias-correct SIC runs into trouble when the bias is so large that sea ice completely disappears over wide areas for too long. Besides this caveat, the choice of CMIP5 AOGCMs used in this study was guided by compliance to desired characteristics of the climate change signal rather than by the skills of the models in reproducing SSC in the historical periods.

Therefore, we identified CMIP5 models with the strongest and weakest climate change signal by the end of the 21st century considering only SSC in the Southern Ocean, in order to span the uncertainty range associated with model response. We computed the relative evolution of integrated winter SIE over the whole Southern Ocean between the historical simulation (reference period: 1971-2000) and the RCP8.5 scenario (reference period: 2071-2100) for 21 AOGCMs from CMIP5 experiment. The CMIP5 ensemble was reduced to 21 because some models sharing the same history of development and high code comparability as others have been discarded. The model list is the same as in Krinner and Flanner (2018) and can be seen in the Fig. 1 legend. We also looked at the mean summer SST increase South of 60°S for the same reference periods. In order to be consistent with periods of maximum (minimum) SIE, seasons considered in this analysis are shifted, and winter (summer) corresponds here to the period August-September-October, ASO (February-March-April, FMA).

The results of the computation can be seen in Fig. 1, which displays the relative late winter (ASO) decrease in SIE in the RCP8.5 projections as a function of the value of the late winter SIE in the historical simulation. The four models with the strongest SIE decrease are CNRM-CM5 (-62.4 %), GISS-E2-H (-53.4 %), inmcm4 (-47.9%) and MIROC-ESM (-45.2 %). Because of the above-mentioned limitation of the bias-correction method, the first three GCMs cannot be selected due to a large negative bias of winter and spring SIE. We therefore selected MIROC-ESM as representative for models projecting a large climate change signal for sea- ice around Antarctica. Conversely, MIROC5 shows the lowest decrease (-1.5%) followed by NorESM1-M (-13,6%). For the same reasons of limitations of the bias correction method, we dismissed MIROC5 and kept NorESM1-M as representative for a weak climate change signals in the SSC around Antarctica. The impact of primarily considering changes in winter SIE rather than in late summer SST is limited as the climate change signal for these two variables are strongly correlated ($R^2$=0.96). For late summer SSTs, MIROC-ESM shows the 6[th] largest increase (+1.8 K), while NorESM1-M exhibits the second lowest (+0.4 K).

## 2.2 CNRM-ARPEGE set-up

We use version 6.2.4 of AGCM ARPEGE, a spectral primitive equation model from Météo-France, CNRM (Déqué et al., 1994). The model is run at T255 truncation with a 2.5 zoom factor and a pole of stretching at 80°S and 90°E. With this setting, the horizontal resolution in Antarctica ranges from 30 km near the stretching pole on the Antarctic Plateau to 45 km at the Northern tip of the Antarctic Peninsula. At the Antipodes, near the North Pole, the horizontal resolution decreases to about 200 km. In this model version, the atmosphere is discretized into 91 sigma-pressure vertical levels. The surface scheme is SURFEX-ISBA-ES (Noilhan and Mahfouf, 1996) which contains a three-layer snow scheme of intermediate complexity (Boone and Etchevers, 2001) that takes into account the evolution of the surface snow albedo, the heat transfer through the snow layers and for the percolation and refreezing of liquid water in the snow pack. Over the ocean, we use a 1D version of sea ice model GELATO (Mélia, 2002) which means that no advection of sea ice is possible. The sea ice thickness is prescribed following the empirical parametrization used in Krinner et al. (1997, 2010) and described in Beaumet et al. (2019). The use of GELATO is therefore limited to the computation of heat and moisture fluxes in sea ice covered regions and also allows taking into account for the accumulation of snow on top of sea ice.

We performed an AMIP-style control simulation for the period 1981-2010 in which CNRM-ARPEGE is driven by observed SST and SIC coming from PCMDI data set (Taylor et al., 2000). CNRM-ARPEGE was also forced by the original oceanic SSC coming from the historical simulations of MIROC-ESM and NorESM1-M (1981-2010) and from projections under the radiative concentration pathway RCP8.5 (Moss et al., 2010) carried out with the same two models (2071-2100). In section 3.2, we present modelled climate at the end of the 21[st] century by CNRM-ARPEGE and the differences in climate change signal between projections realized with bias-corrected and original SSC from MIROC-ESM and NorESM1-M RCP8.5.

In each ARPEGE simulation, the first two years are considered as a spin-up phase for the atmosphere and the soil or snowpack, and are therefore discarded from the analysis. The characteristics of the different ARPEGE simulations presented in this paper are summarized in table1.

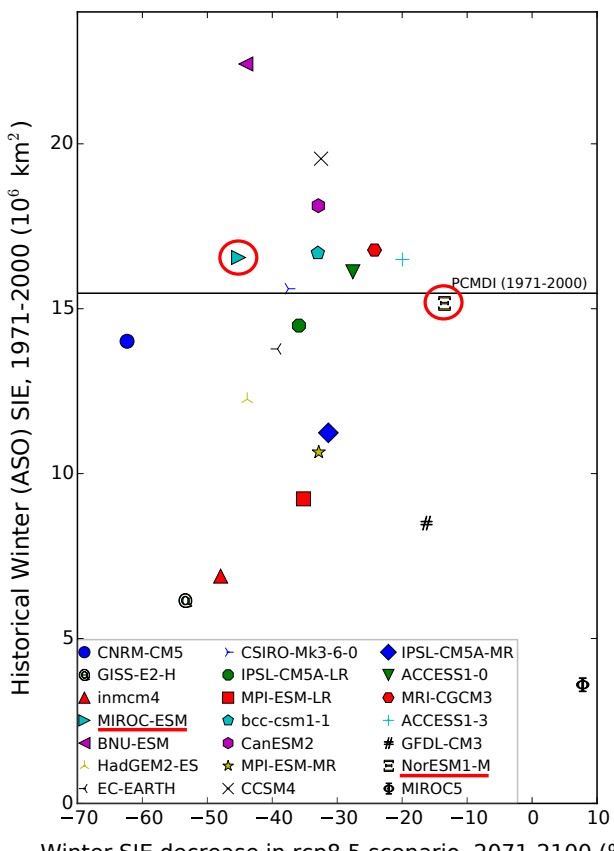

**Figure 1.** Historical Antarctic winter (August-September-October: ASO) sea ice extent (SIE, in millions of km$^2$) as function of the relative decrease of winter SIE in the RCP8.5 projection for the period 2071-2100 with respect to the reference period 1971-2000. The mean winter SIE in the observations for the historical reference period is indicated by the horizontal black line (PCMDI 1971-2000). Model selected for this study are highlighted in red.

**Table 1.** Summary of the period, sea surface conditions, greenhouse gazes (GHG) concentration and reference historical simulation (for climate projections) for each ARPEGE simulation presented in this paper

| Simulations | Period | SSC | GHG Concentrations | Reference for hist. climate |
|---|---|---|---|---|
| ARP-AMIP | 1981-2010 | Observed | historical | - |
| ARP-NOR-20 | 1981-2010 | NorESM1-M historical | historical | - |
| ARP-MIR-20 | 1981-2010 | MIROC-ESM historical | historical | - |
| ARP-NOR-21 | 2071-2100 | NorESM1-M RCP8.5 | RCP8.5 | ARP-NOR-20 |
| ARP-MIR-21 | 2071-2100 | MIROC-ESM RCP8.5 | RCP8.5 | ARP-MIR-20 |
| ARP-NOR-21-OC | 2071-2100 | Bias-corrected NorESM1-M RCP8.5 | RCP8.5 | ARP-AMIP |
| ARP-MIR-21-OC | 2071-2100 | Bias-corrected MIROC-ESM RCP8.5 | RCP8.5 | ARP-AMIP |

## 2.3 Model Evaluation

The ability of ARPEGE model to reproduce atmospheric general circulation of the Southern Hemisphere is assessed by comparing sea level pressure (SLP) and 500 hPa geopotential height (Z500) poleward of 20°S to those of ERA-Interim reanalysis (ERA-I). For surface climate of the Antarctic continent, several studies have shown that (near)-surface temperatures from ERA-I are not reliable (Bracegirdle and Marshall, 2012; Jones and Harpham, 2013; Fréville et al., 2014), as the reanalysis is not constrained by a sufficient number of observations and because the boundary layer physics of the model fails to successfully reproduce strong temperature inversions near the surface that characterize the climate of the EAP. As a consequence, near-surface temperatures in Antarctica from ARPEGE simulations are evaluated using observations from the SCAR READER data base (Turner et al., 2004) as well as temperatures from a MAR RCM simulation in order to increase the spatial coverage of the model evaluation. MAR (Gallée and Schayes, 1994) has been one of the most successful RCMs in reproducing the surface climate of large ice sheet such as Greenland (Fettweis et al., 2005; Lefebre et al., 2005) and Antarctica (Gallée et al., 2015; Amory et al., 2015; Agosta et al., 2018). For Antarctica, outputs of the MAR simulation (version 3.6 of the model) driven by ERA-I have been evaluated against *in-situ* observations for surface pressure, 2 m temperatures, 10 m wind speed and surface mass balance in Agosta et al. (2018) and Agosta (2018). MAR skills for temperatures and SMB are excellent for most of Antarctica. However, a systematic 3-5 K cold bias over large ice shelves (Ross and Ronne-Filchner) throughout the year and a 2.5 K warm bias over the Antarctic Plateau in winter are worth mentioning.

In this evaluation, we compare ARPEGE near-surface temperatures, to those of an ERA-I driven MAR simulation (hereafter MAR-ERA-I) at a similar horizontal resolution of 35 kilometres (Agosta et al., 2018). The SMB of the grounded AIS and its components from ARPEGE simulations are compared to outputs of the same ERA-Interim driven MAR simulation from Agosta et al. (2018). We also performed an evaluation of ARPEGE snowfall rates using a model independent data set such as the CloudSAT climatology for Antarctic snowfall (Palerme et al., 2014). However, because this data set is only available for a very short period of time (2007-2010) and is representative of snowfall rates about 1200 m above the surface, the results from this comparison have to be considered with extreme caution and are therefore only shown in the supplementary material (see C2). In this manuscript, the statistical significance of the differences is assessed using a double-sided t-test and at the 5% level.

## 3 Results

### 3.1 Simulated Present Climate

In this section, ARPEGE simulation are evaluated using mostly ERA-I reanalyses for atmospheric general circulation south of 20°S and polar-oriented RCMs as well as READER *in-situ* data for the surface climate of the ice sheet.

#### 3.1.1 Atmospheric General Circulation

The differences between mean SLP from the 1981-2010 ARPEGE simulation driven by observed SSC (called ARP-AMIP in the remainder of this paper, see Table 1) and mean SLP from ERA-I reanalysis can be seen in Fig. 2a. The general pattern

**Table 2.** Seasonal root mean square error (RMSE, in hPa) on mean SLP South of 20°S with respect to ERA-Interim for the different ARPEGE simulations over the 1981-2010 period. Each error is significant at p=0.05

| Simulations | DJF | MAM | JJA | SON |
|-------------|-----|-----|-----|-----|
| ARP-AMIP | 3.3 | 2.7 | 3.1 | 3.0 |
| ARP-NOR-20 | 3.5 | 4.3 | 4.8 | 4.6 |
| ARP-MIR-20 | 3.2 | 4.0 | 4.6 | 3.2 |

is an underestimation of SLP around 40°S, especially in the Pacific sector (up to 6 to 10 hPa) and an overestimation around Antarctica (generally between 4 and 8 hPa), especially in Amundsen/Ross sea sector. Mean SLP differences for ARPEGE simulations driven by NorESM1-M (ARP-NOR-20) and MIROC-ESM (ARP-MIR-20) historical SSC can be seen respectively in Fig. 2b and Fig. 2c. The pattern and the magnitude of the errors are similar to those of the ARP-AMIP simulation in summer (DJF). The seasonal root mean square errors (RMSE) for each simulation are summarized in Table 2. In winter (JJA), spring (SON) and autumn (MAM) the errors are substantially larger in ARP-NOR-20 and ARP-MIR-20 than in ARP-AMIP (up to 50% larger). The patterns of the errors and the ranking of simulation scores are similar for the 500hPa geopotential height(*not shown*) and SLP.

The mean atmospheric general circulation in each simulation has also been compared and evaluated against ERA-I by analyzing the latitudinal profile of the 850 hPa zonal mean eastward wind component (referred to as westerly winds in the following), as well as the strength (m/s) and position (°Southern latitude) of the zonal mean westerly wind maximum (Fig. 3). In this figure, results are only presented for the annual average, as the differences between simulations or with respect to ERA-I do not depend much on the season considered (*not shown*). ARP-AMIP and ARP-MIR-20 are closer to ERA-I when the westerly winds maximum strength is considered with an underestimation of this maximum about 1.5 m.s$^{-1}$s. The equatorward bias on the position of the westerly wind maximum is 1.6°in ARP-NOR-20, while it is up to 3 to 5°in ARP-AMIP and ARP-MIR-20.

### 3.1.2 Near-surface Temperatures

Screen level (2 m) air temperatures (T$_{2m}$) from ARP-AMIP simulation are compared to those from MAR-ERA-I simulation and READER data base in winter (JJA) and summer (DJF) for the reference period 1981-2010 (Fig. 4). In this analysis, stations from the READER data base for which less than 80% of valid observations were recorded for the reference period were not used for the computation of the climatological mean. Altitude differences between corresponding ARPEGE grid point and stations have been accounted for by correcting modelled temperatures with a 9.8 K km$^{-1}$ dry adiabatic lapse rate, such as done for instance in Bracegirdle and Marshall (2012). Errors of the T$_{2m}$ in ARP-AMIP simulation for each weather station and each season are presented in the supplementary material (Table B1).

The ARP-AMIP T$_{2m}$ are much warmer than MAR-ERA-I on the ridge and the western part of the Antarctic Plateau in winter as well as on on the large Ronne and Ross ice shelves. Consistently with its atmospheric circulation errors in this area, ARPEGE is colder than MAR-ERA-I on the Southern and Western part of the Antarctic Peninsula, especially in winter. We

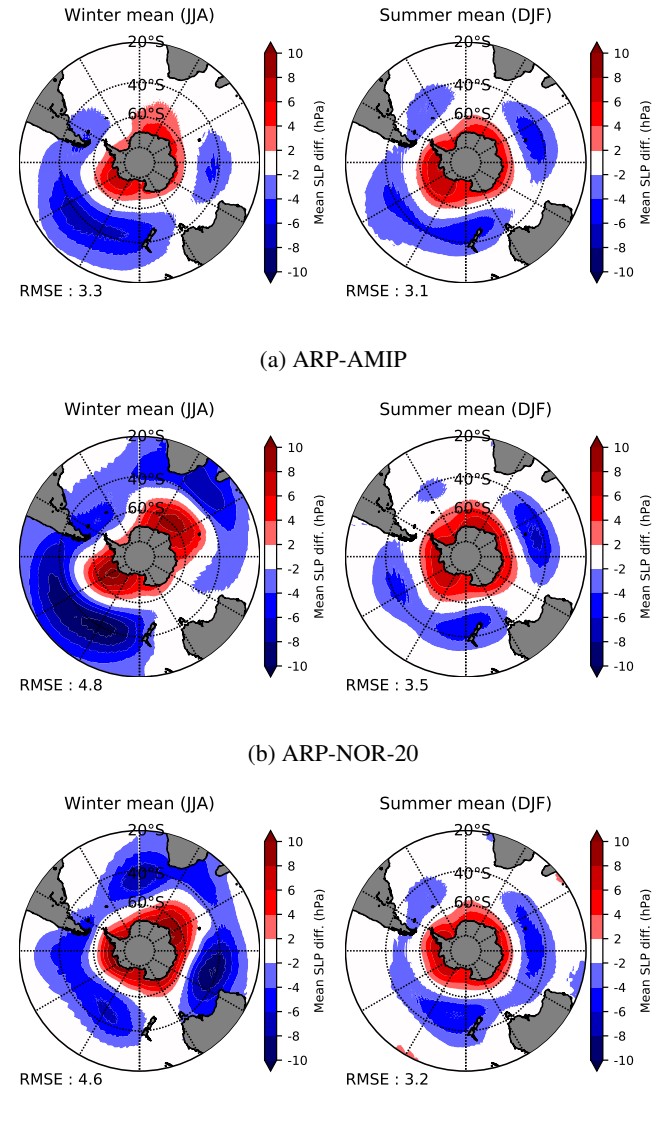

**Figure 2.** Difference between ARPEGE simulations and ERA-I mean SLP for the reference period 1981-2010 in winter (JJA) and summer (DJF). Value of the RMSE are given below the plots.

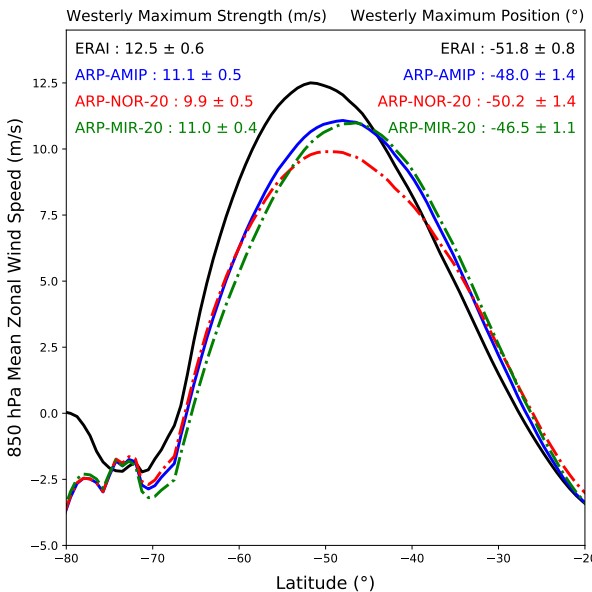

**Figure 3.** Mean latitudinal profile of 850 hPa eastwards wind component (reference period : 1981-2010) for ARP-AMIP (grey), ARP-MIR-20 (dashed green), ARP-NOR-20 (dashed red) and ERA-Interim (black). Yearly mean $\pm$ one standard deviation of strength (m.s$^{-1}$, upper left) and latitude position ($°$, upper right) of the 850 hPa westerly wind maximum.

can also mention a moderate (1 to 3 K) but widespread warm bias on the slope of the EAP and on the west side of the West Antarctic Ice Sheet (WAIS) in summer. Except for some coastal stations of East Antarctica, $T_{2m}$ errors in the ARP-AMIP simulation are very similar in the comparisons with MAR-ERA-I and READER data base.

Considering errors on near-surface temperatures of the Antarctic Plateau as large as 3 to 6 K for ERA-I reanalysis in all seasons (Fréville et al., 2014), skills of the ARP-AMIP simulation in this region are comparable to those of many AGCM or even climate reanalyses. The systematic error for Amundsen Scott station is for instance not significant at the 5% level in any season except autumn (MAM). The large discrepancies between ARPEGE and MAR over large ice shelves are further investigated in the appendix B1. Although a part (3-5K) of this large discrepancy in winter (ARPEGE up to 12 K warmer than MAR over the center of Ross Ice Shelf) comes from a cold bias in MAR identified in the comparison with the in-situ observations (Agosta, 2018), the majority of ARPEGE errors on large ice shelves appears to come from specificities in the representation of stable boundary layers over these large and flat surfaces. As a consequence, the surface climate over the large ice shelves simulated by ARPEGE should at this stage be used with circumspection. Considering the model lower skills on the floating ice shelves, integrated SMB and temperature changes are mostly presented and discussed for the grounded ice sheet (GIS) in the remainder of the paper.

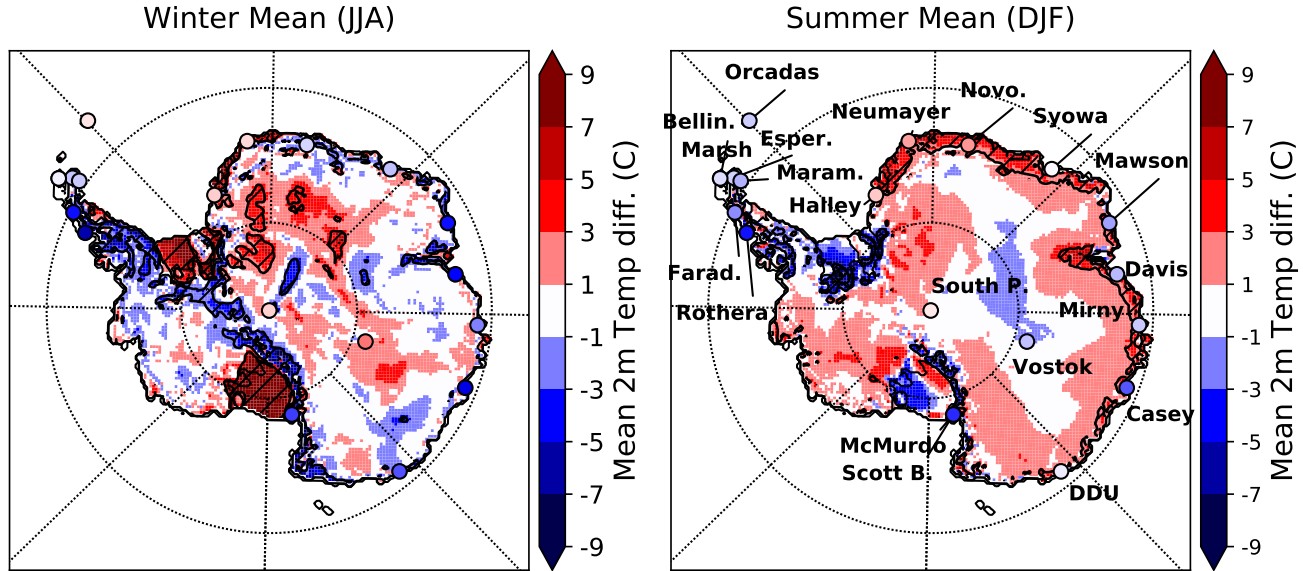

**Figure 4.** $T_{2m}$ differences between ARP-AMIP and MAR-ERA-I (Agosta et al., 2018) simulations in winter (JJA, *left*) and summer (DJF, *right*) for the reference period 1981-2010. Circles are $T_{2m}$ differences between ARP-AMIP and weather stations from the READER data base, stations names are shown on the right side pannel ("Bellin." = Bellingshausen,"DDU"= Dumont D'Urville, "Esper." = Esperanza, "Farad." = Faraday, "Maram." = Marambio, "Novo." = Novolerevskaya, "South P." = South Pole-Amundsen Scott). Black hatched areas is where $|\,\mathrm{ARPEGE} - \mathrm{MAR}\,| = 1\mathrm{MAR}\sigma$.

Large negative biases in ARP-AMIP for some coastal stations of East Antarctica (Casey, Davis, Mawson, Mc Murdo), especially in winter, are likely due to effects of the local topography that cannot be captured at a 35 kms horizontal resolution. Besides, ARPEGE temperatures are representative for a 35x35 km² inland grid point, whereas many weather stations are located very close to the shoreline. The large cold bias at Rothera station on the Peninsula is likely a combination of the effects

5 of poorly represented local topography in the model and of errors on the simulated atmospheric general circulation.

Regarding $T_{2m}$ in ARPEGE simulations forced by NorESM1-M and MIROC-ESM historical SSC, the skills of the ARPEGE model are particularly impacted over the AP and, to a lesser extent, over the EAP (see Fig. B1). Over coastal East Antarctic stations, most of the errors in $T_{2m}$ are likely due to local topography effects, or inadequacies of the physics of the atmospheric model, as the skills of the atmospheric model shows few variations in the three simulations. The use of SSC from NorESM1-M

10 and MIROC-ESM instead of observed SSC also impacts the simulated temperatures at the continental scale. Differences for ARP-NOR-20 and ARP-MIR-20 in $T_{2m}$ for the Antarctic GIS with respect to the ARP-AMIP simulation are presented in Tab. 3. For the ARP-MIR-20, differences of -0.7 K in spring and -1.5 K in summer were found significant. For ARP-NOR-20, differences ranging from 0.4 K to 1.2 K in autumn, winter and spring are significant as well.

**Table 3.** Mean seasonal $T_{2m}$ differences (in K) for the GIS with respect to the ARP-AMIP simulation. Differences significant at p=0.05 are presented in bold.

| Simulations | DJF | MAM | JJA | SON |
|---|---|---|---|---|
| ARP-NOR-20 | -0.1 | **0.4** | **1.2** | **0.9** |
| ARP-MIR-20 | **-1.5** | -0.2 | 0.3 | **-0.7** |

### 3.1.3 Surface Mass Balance

In this study, SMB from ARPEGE simulations is defined as the total precipitation minus the surface snow sublimation/evaporation minus the surface run-off. Differences between ARP-AMIP and MAR-ERA-I total precipitation, snow sublimation and SMB (in mm of water equivalent per year) for the reference period 1981-2010 can be seen in Fig. 5. As differences in runoff are

restricted to the ice shelves and some very localized coastal areas, their spatial distribution is not displayed in this figure. Yearly mean SMB, total precipitation, sublimation, run-off, rainfall and melt, integrated over the whole Antarctic GIS for the different ARPEGE simulations, for MAR and RACMO2 driven by ERA-Interim reanalyses and from other studies are presented in Table 4.

Precipitation integrated over the grounded ice sheet in ARP-AMIP ad ARP-MIR-20 is very close to the values from MAR-
ERA-I and RACMO2-ERA-I. However, higher surface sublimation (and run-off) in ARPEGE simulation tend to yield lower estimates of the GIS integrated SMB. Integrated SMB over the ice sheet using ARPEGE however concurs independent estimates from satellite data (e.g., Vaughan et al., 1999; Arthern et al., 2006). Precipitation is generally much higher in ARPEGE with respect to MAR over many coastal areas such as the Ross sector of Marie Byrd Land, in Dronning Maud and in the northern and eastern part of the AP. On the other hand, precipitation is lower in ARP-AMIP in the western part of the Peninsula, in
the inland part of central WAIS and in the interior and lee-side of the TransAntarctic Mountains. Sublimation integrated over the grounded ice sheet is about three times higher in ARP-AMIP than in MAR-ERA-I. Differences mostly come from coastal areas and the peripheral ice sheet. This is consistent with ARP-AMIP being systematically 1 to 3 K warmer than MAR-ERA-I in summer in those areas. The inter-annual variability is very high in the simulated ARPEGE runoff, and so it is in MAR-ERA-I. A closer look at the values of rainfall, surface snow melt and runoff in the three present-day ARPEGE simulations in Table 4
shows that about 1/3 of the liquid water input into the snowpack (rainfall + surface snow melt) does not refreeze and therefore leaves the snowpack in the end. In MAR-ERA-I and in RACMO2-ERA-I, this ratio is about 1/20. This means that although the snow surface scheme SURFEX-ISBA used in ARPEGE is in principle able to explicitly account for storage and refreezing of liquid water in the snow-pack, the retention capacity of the Antarctic snow-pack appears to be largely underestimated when compared to MAR and RACMO2. For these reasons, projected changes in melt rates are preferably presented and discussed
in section 3.2, while changes in run-off are *not shown* due the suspected lower skill of ARPEGE for this variable and strong non-linearities generally expected in changes in surface run-offs in a warming climate.

| Simulation | SMB | Precip. | Subli. | Run-Off | Rain | Melt |
|---|---|---|---|---|---|---|
| ARP-AMIP | 1970±96 | 2268±94 | 277±17 | 22±14 | 10±2 | 52±32 |
| ARP-NOR-20 | **2188±101** | **2484±100** | 275±12 | 21±14 | 10±2 | 52±27 |
| ARP-MIR-20 | 1996±84 | 2267±92 | **257±18** | **14±9** | 10±3 | **34±21** |
| MAR-ERA-I[1] | 2158±106 | 2260±104 | 84±10 | 3±2 | 16±3 | 45±15 |
| RACMO2-ERA-I[1] | 2117±92 | 2268±99 | 136±4 | 2±2 | 3±1 | 61±21 |
| RACMO2-ERA-I[2](entire ice sheet) | 2596±121 | 2835±122 | 228±11 | 5±2 | 6±2 | 88±24 |
| CESM-hist[3] | 2280±131 | 2433±135 | 68±6 | 86±21 | 5±2 | 203±41 |
| Vaughan et al. (1999) | 1811 | | | | | |

**Table 4.** Mean Grounded AIS SMB and its component (Gt yr$^{-1}$) ± one standard deviation of the annual mean for the reference period 1981-2010. Variables from ARP-NOR-20 and ARP-MIR-20 that are significantly different from the value in ARP-AMIP at p=0.05 level are in **bold**. [1]MAR and RACMO2 driven by ERA-I and ARPEGE statistics for 1981-2010 over the Antarctic GIS are computed using MAR grounded ice mask (area = 12.37 10$^6$ km$^2$) such as in Agosta et al. (2018). Sublimation values for RACMO2 include drifting snow sublimation, while only surface sublimation is accounted in MAR and ARPEGE statistics.[2]RACMO2 statistics are given for the total Ice Sheet and the period 1979-2005 from Lenaerts et al. (2016), sublimation includes drifting snow sublimation. [3]Community Earth System Model historical simulation (1979-2005), values for the total ice-sheet from Lenaerts et al. (2016)

In the ARP-MIR-20 simulation, snow sublimation, run-off and melt were found significantly lower than in ARP-AMIP, which is consistent with this simulation being 1.5 K cooler in summer (DJF). The effect of driving ARPEGE by biased SSC for the modelling of Antarctic precipitation is discussed in the supplementary material (see Sec. C1).

## 3.2 Climate change signal

In this section, we present the climate change signal obtained in ARPEGE RCP8.5 projections driven by SSC from NorESM1-M and MIROC-ESM. For ARPEGE projections realized using original SSC from the two coupled models (ARP-NOR-21 and ARP-MIR-21), the reference simulations for the historical period are the ARPEGE simulations performed with historical SSC coming from the respective coupled model (ARP-NOR-20 and ARP-MIR-20). For scenarios realized with bias-corrected SSC (ARP-NOR-21-OC and ARP-MIR-21-OC), the reference simulation for the historical period is ARP-AMIP (observed SSC). The primary goal here is to evaluate the effect in climate change signals for Antarctica associated with oceanic forcings coming from the end valued of the CMIP5 ensemble in terms of sea ice retreat and the changes coming from the bias correction of the SSC.

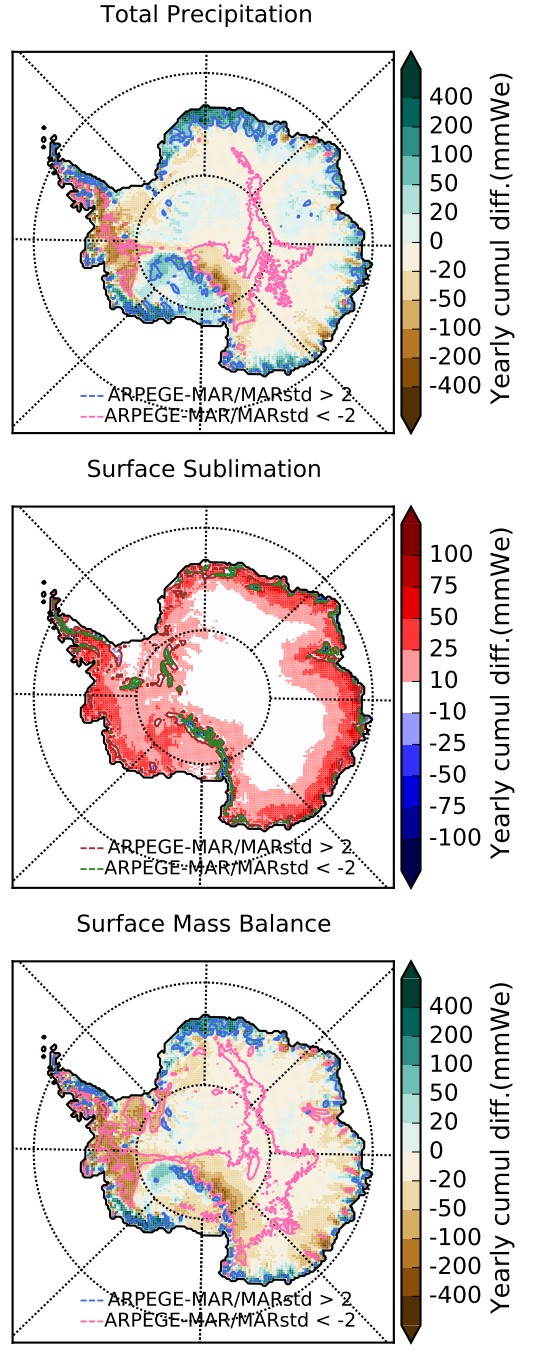

**Figure 5.** Total precipitation (*top*), Sublimation/Evaporation (*centre*) and SMB (*bottom*) for ARP-AMIP minus MAR-ERA-I difference (mm.we yr$^{-1}$) for the reference period 1981-2010. Pink (brown) and blue (green) contour lines represents areas where ARPEGE-MAR differences are respectively smaller than -2 or bigger than 2 MAR standard deviation of annual mean ($2\sigma$).

**Table 5.** Changes in mean yearly Southern westerly wind maximum strength ($\Delta$JSTR, m/s) and position ($\Delta$JPOS, °) for the different ARPEGE projections. Changes significantly different using bias-corrected SSC are shown in **bold**.

| Simulations | $\Delta$JSTR (m/s) | $\Delta$JPOS (°) |
|---|---|---|
| ARP-NOR-21 | 1.7 | -0.8 |
| ARP-NOR-21-OC | 1.5 | **-2.2** |
| ARP-MIR-21 | 1.9 | -3.7 |
| ARP-MIR-21-OC | 2.0 | -3.8 |

### 3.2.1 Atmospheric General Circulation

Climate change signals in mean SLP for the different RCP8.5 projections realized with ARPEGE can be seen in Fig. 6. Each one shows a pressure increase at mid-latitudes (30-50 °S) and a decrease around Antarctica. This corresponds to a strengthening of the mid to high latitude pressure gradient (positive phase of the SAM) and a poleward shift of the circum-Antarctic low
pressure belt towards the continent, which are generally the expected consequences of 21$^{\text{st}}$ century climate forcing (Kushner et al., 2001; Arblaster and Meehl, 2006). This pattern (increase at mid-latitude, decrease around Antarctica) is sharper in projections realized with MIROC-ESM SSC.

Differences in the climate change signal for ARP-NOR-21-OC and in ARP-NOR-21 with respect to their corresponding references in historical climate are smaller. Differences in SLP changes are larger in the projections realized with MIROC-ESM
SSC : in those with non bias-corrected SSC (ARP-MIR-21), the intensification of the low pressure systems around Antarctica in winter is clearly organized in a 3-wave pattern (Fig. 6b). In ARP-MIR-21-OC, the JJA pressure decrease is rather organized in a dipole with one maximum of pressure decrease centered the eastern side of the Ross Sea and the other west of the Weddell Sea. As a result, the 3-wave pattern is clearly noticeable in the difference between the two climate change signals (Fig. 6b, *right*). Late 21$^{\text{st}}$ century changes in westerly wind maximum latitude position and strength at 850 hPa are shown in Table 5.
When compared to the variability in the reference historical simulations, each climate change signal is significant at the 5% level. Regarding the changes in westerly winds maximum strength, the difference between the two projection using NorESM1-M SSC are limited. However, we can mention a 1.4° stronger southward displacement of the westerly wind maximum position in the projection using bias-corrected SSC (significant at the 5% level). Differences in changes in position and strength are not significant between ARP-MIR-21 and ARP-MIR-21-OC. Compared to projections realized with SSC from NorESM1-M,
these projections show a slightly larger increase in westerlies maximum strength and a much larger poleward shift, although this difference is reduced when comparing projections with bias-corrected SSC.

### 3.2.2 Near-surface temperatures

The mean yearly $T_{2m}$ increase for the Antarctic GIS using SSC from NorESM1-M RCP8.5 projection is 2.9$\pm$1.0 K using original SSC (ARP-NOR-21) and 2.8$\pm$0.8 K using bias-corrected SSC (ARP-NOR-21-OC). For scenarios using SSC from
MIROC-ESM, these temperatures increases are respectively 3.8$\pm$0.7 K and 4.2$\pm$1.0 K. The differences in yearly $T_{2m}$ increase

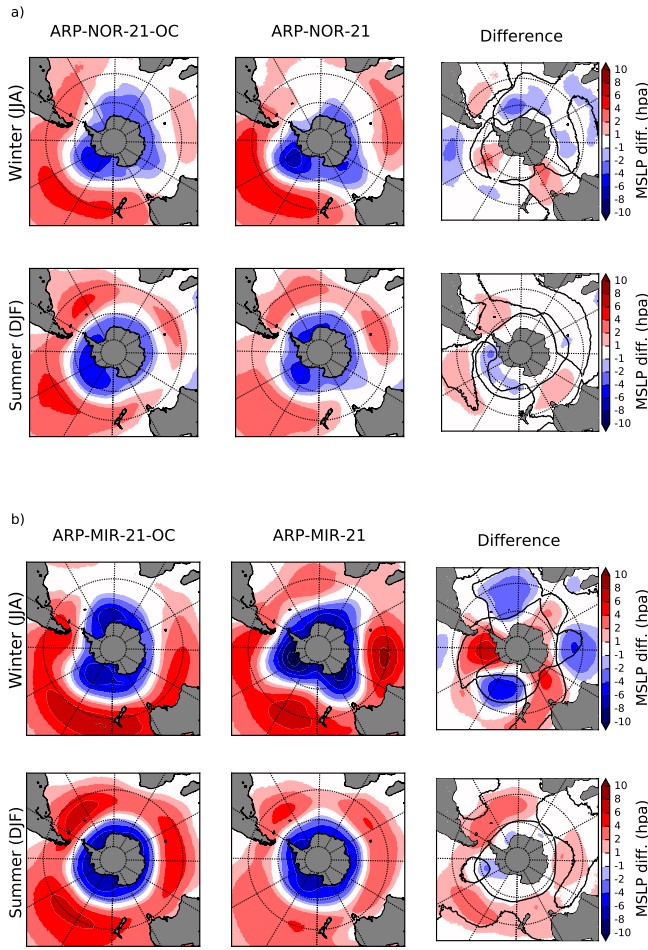

**Figure 6.** Climate change signal in SLP for ARPEGE RCP8.5 projections with bias corrected SSC (*left*), original SSC (*center*) and difference (*right*). Climate change signal for winter (JJA) are displayed at the *top* of the subfigures and for summer (DJF) at the *bottom*. Results for scenarios with SSC from NorESM1-M are presented in upper (a) and from MIROC-ESM in lower (b) part of the figure. *Black contour lines* represent areas where differences in climate signal is 50% of the climate signal in the simulation with non bias-corrected SSC.

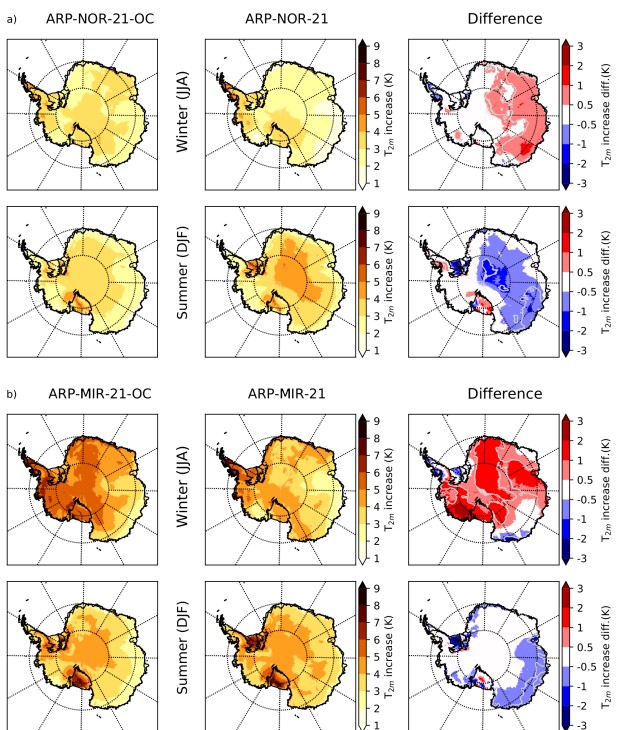

**Figure 7.** Climate change signal in $T_{2m}$ for ARPEGE RCP8.5 projections for the late 21$^{st}$ century (2071-2100) with bias-corrected SSC (*left*), original SSC (*center*) and difference (*right*). Climate change signal for austral winter (summer) are displayed at the *upper* (*lower*) part of the figure. Results for projections with SSC from NorESM1-M are presented in (a) and from MIROC-ESM in (b). *Grey contour lines* is where differences in climate change signal is 25% of the climate change signal using non bias-corrected SSC

using bias-corrected SSC are found non significant in both cases. $T_{2m}$ increase per season can be seen in Tab. 6. Only a +0.8 K difference in winter temperatures increase in ARP-MIR-21-OC with respect to the projection driven by original SSC is found significant. At the regional scale (Fig. 7b), this is materialized by large areas of 1 to 2 K stronger warming in the centre of the East Antarctic Plateau, Dronning Maud Land and the Ross Ice Shelf. The difference in warming in ARP-MIR-21-OC is the highest in Marie-Byrd Land (+2 K).

For projections using SSC from NorESM1-M, no seasonal differences were found significant at the AIS scale.

### 3.2.3 Precipitation and Surface Mass Balance

Absolute values and changes in Antarctic GIS SMB and its components for the late 21$^{st}$ are shown in Table 7. For the experiment realized with NorESM1-M SSC, precipitation and SMB changes (in both cases increases) are very similar (no significant differences), while there is about 220 Gt.yr$^{-1}$ more precipitation and therefore accumulation in ARP-NOR-21 absolute values (significant at p=0.05). No significant differences in absolute values or climate change signals were found for the other components of SMB for scenarios with NorESM1-M SSC.

**Table 6.** Mean season $T_{2m}$ increase (K) for the Antarctic GIS for the different ARPEGE RCP8.5 scenario at the end of $21^{st}$ century (reference period: 2071-2100) with respect to their historical reference simulation (reference period: 2071-2100). Climate change signal in scenarios with bias-corrected SSC significantly different at p=0.05 level are presented in bold.

| Simulations | DJF | MAM | JJA | SON |
|---|---|---|---|---|
| ARP-NOR-21 | 3.5±1.4 | 2.7±1.4 | 2.6±2.0 | 2.7±1.4 |
| ARP-NOR-21-OC | 3.0±1.4 | 2.6±1.4 | 3.1±1.4 | 2.6±1.0 |
| ARP-MIR-21 | 3.9±0.9 | 4.1±1.3 | 3.8±1.4 | 3.5±1.2 |
| ARP-MIR-21-OC | 3.6±1.5 | 4.6±1.7 | **4.6±1.4** | 3.8±1.5 |

For the experiment performed with MIROC-ESM SSC, absolute values and increase in precipitation are about 170 Gt.yr$^{-1}$ (7 %) stronger in the projection with bias-corrected SSC. The total precipitation increase is +8.8% K$^{-1}$ in ARP-MIR-21-OC, compared to a 7.9% K$^{-1}$ increase in ARP-MIR-21. For SMB and precipitation, both absolute values and climate changes signals were found significantly different in ARP-MIR-21-OC than in ARP-MIR-21.

In each projection, the sublimation increases by about 20 to 30% with respect to the corresponding values in the historical period. Surface melt increases by about a factor 2 to 3 in scenarios with NorESM1-M SSC and by factors from 5 to 6 in projections with MIROC-ESM SSC. Increases in SMB remain essentially determined by the increases in precipitation. As a consequence, we only present here the spatial distribution of changes in precipitation in Antarctica in Fig. 8. In all projections, the strongest absolute precipitation increases occur in the coastal regions of West Antarctica and in the west of the Peninsula. In

simulations with MIROC-ESM SSC, precipitation increase is also very strong in the Atlantic sector of coastal East Antarctica. The difference between total precipitation increases in ARP-NOR-21 and ARP-NOR-21-OC (Fig. 8a) is small in most regions of Antarctica, except for a stronger increase (or weaker decrease) in Marie-Byrd Land, and a weaker increase in Adélie Land in ARP-NOR-21-OC. For the simulations with MIROC-ESM SSC (Fig. 8b), we can clearly identify an alternation of three regions of higher or lower precipitation increases. This tri-pole pattern can easily be linked to the 3-wave pattern in SLP change in

ARP-MIR-21, clearly different than the pattern in MSLP change in ARP-MIR-21-OC (Fig. 6b). Here again, Marie Byrd Land and Adélie Land are among the areas where large differences are found between simulations with or without bias-corrected SSC. Winter and spring (and to a lesser extent autumn) are the seasons mostly responsible for differences in precipitation changes between the simulations with MIROC-ESM original SSC. The relative mean precipitation changes (in %) and the associated standard deviation for the four RCP8.5 projections realized in this study can be seen in Fig. 9.

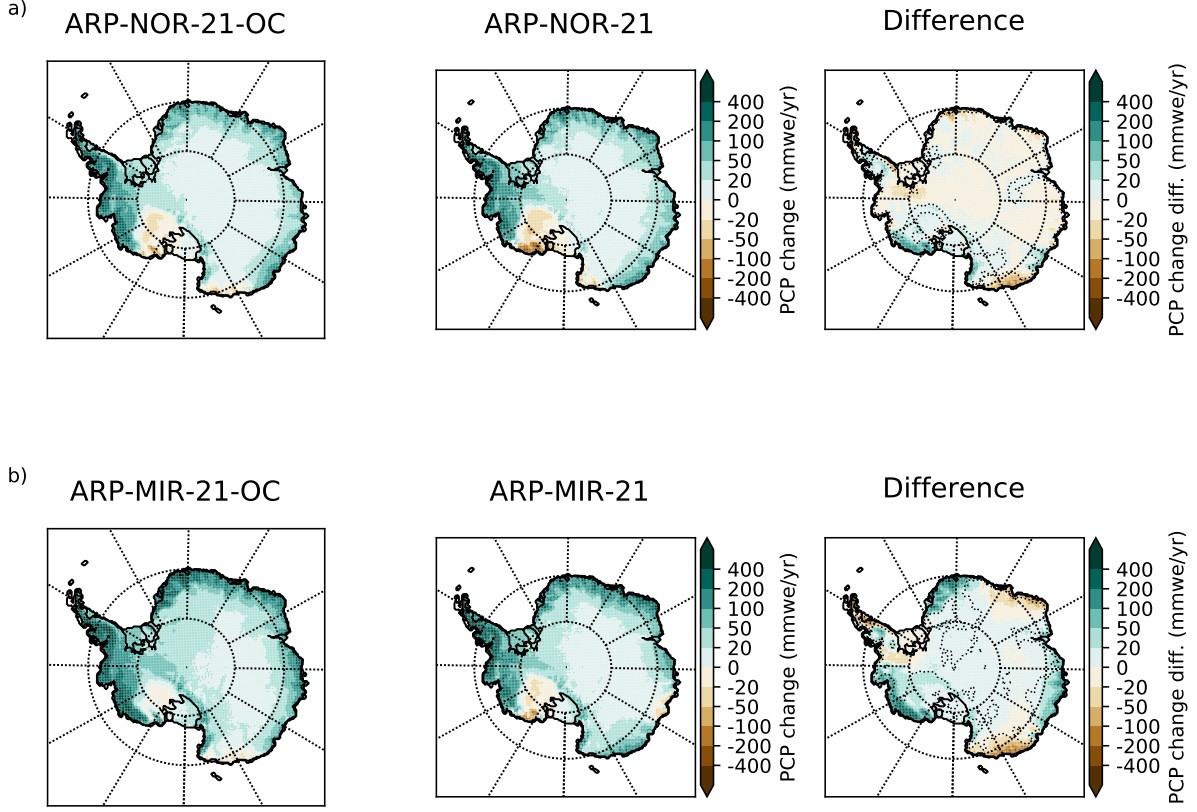

**Figure 8.** Climate change signal in total precipitation for late $21^{st}$ century (reference period: 2071-2100) in ARPEGE RCP8.5 projection with bias corrected SSC (*left*), original SSC (*center*) and difference (*right*). Results for scenarios with SSC from NorESM1-M are presented in subfigure (a) and from MIROC-ESM in subfigure (b). *Dotted lines* indicate where difference is 50% of the precipitation change in the non bias-corrected SSC projection.

**Table 7.** Absolute values, absolute (Gt yr$^{-1}$) and relative climate change signal (in %) for Mean SMB and components for the Antarctic GIS for the different ARPEGE RCP8.5 projection (2071-2100). Climate change signals and absolute values significantly different at p=0.05 level in projections with bias-corrected SSC are displayed in bold.

| Simulations | SMB | Tot. PCP | Surf. Sublim. | Rainfall | Melt |
|---|---|---|---|---|---|
| **ARP-NOR-21** | 2543±143 | 2965±167 | 340±28 | 26±6 | 196±102 |
| *CC change (Gt yr$^{-1}$)* | 355±196 | 481±196 | 65±26 | 16±8 | 144±81 |
| *Rel. change* | 16% | 19% | 24% | 164% | 276% |
| **ARP-NOR-21-OC** | **2334±181** | **2742±176** | 331±21 | 27±7 | 184±82 |
| *CC change (Gt yr$^{-1}$)* | 364±195 | 474±179 | 55±26 | 17±8 | 132±137 |
| *Rel. change* | 19% | 21% | 20% | 171% | 252% |
| **ARP-MIR-21** | 2508±98 | 2940±131 | 332±24 | 46±12 | 248±120 |
| *CC change (Gt yr$^{-1}$)* | 512±132 | 673±135 | 75±18 | 31±10 | 248±120 |
| *Rel. change (%)* | 26% | 30% | 29% | 377% | 628% |
| **ARP-MIR-21-OC** | **2637±156** | **3108±202** | 345±29 | 52±15 | 306±144 |
| *CC change (Gt yr$^{-1}$)* | **667±202** | **840±227** | 68±23 | 42±15 | 254±118 |
| *Rel. change* | 34% | 37% | 25% | 416% | 484% |

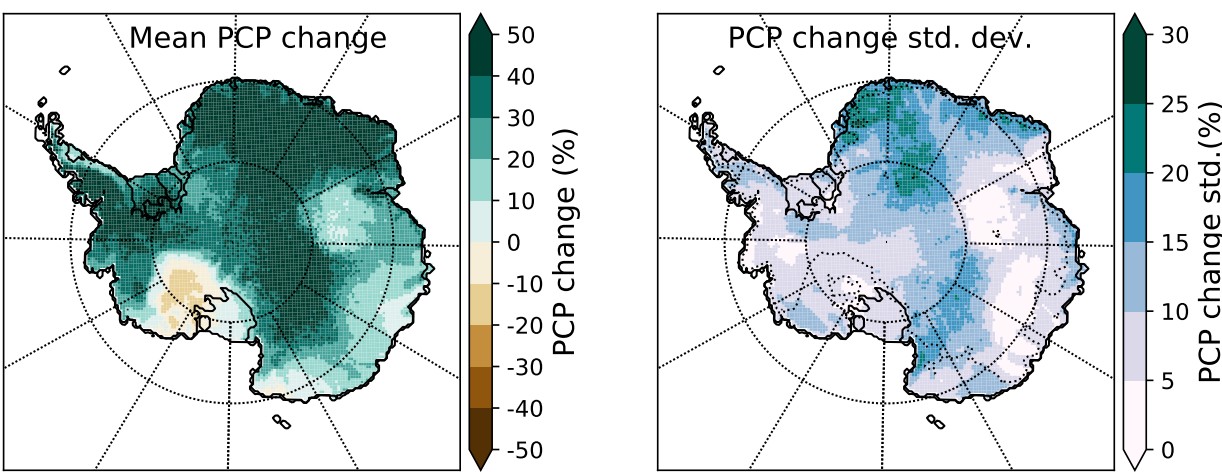

**Figure 9.** Mean (*left*) relative precipitation change (%) for late 21$^{\text{st}}$ century from the four ARPEGE RCP8.5 projections and associated standard deviation (*right*). *Dotted lines* indicate where standard deviation is 50% of the mean change.

## 4 Discussion

### 4.1 Evaluation of ARPEGE climate model : reconstruction of historical climate

The atmospheric model ARPEGE correctly captures the main features of the atmospheric circulation around Antarctica. The three local minima in SLP and 500 hPa geopotential heigh located around 60°W, 90°E and 180 °E are well reproduced in the ARP-AMIP simulation (see Fig. D1a). However, there is a positive SLP bias in the seas around Antarctica, particularly in the ASL sector, and a negative bias at mid-latitudes (30-40 °S), especially in the Pacific sector. This bias structure in the Southern Hemisphere is present in many coupled and atmosphere-only GCMs. Its consequence is an equatorward bias on the position of the surface jet associated with westerly winds (Bracegirdle et al., 2013). The errors of our high resolution ARPEGE on atmospheric general circulation in the high southern latitudes are typical of many lower resolution climate simulation and in the same order of values as the errors of the CMIP5 CNRM-CM5 and ARPEGE (AMIP) simulations found in Bracegirdle et al. (2013). Even though simulations realized with different versions of the model are to be compared with care, our results suggest that here the use of higher resolution did not improved the representation of the high southern latitude atmospheric circulation, contrary to the results of Hourdin et al. (2013) who used LMDZ model.

The use of observed SSC (ARP-AMIP) rather than SSC from NorESM1-M and MIROC-ESM substantially improves the simulated mean SLP in the Southern Hemisphere in all seasons but summer. This confirms at a higher resolution results from previous studies realized at coarser resolution which have shown that the use of observed rather than modeled SSC to drive atmosphere-only model clearly improves the skill of the atmospheric models (Krinner et al., 2008; Ashfaq et al., 2011; Hernández-Díaz et al., 2017).

Regarding surface climate, ARPEGE also reasonably reproduces Antarctic $T_{2m}$ except over large ice shelves. The $T_{2m}$ errors with respect to MAR-ERA-I are generally below 3 K over most of the GIS. There is a substantial warm bias on the top the Antarctic Plateau in winter. However, these errors (+1.5 K at Amundsen-Scott, +3.4 K at Vostok) are to be compared with errors sometimes much larger in other GCMs or even in reanalyses (e.g. Bracegirdle and Marshall, 2012; Fréville et al., 2014). These errors are due to the fact that many climate models fail to capture the strength of the near-surface temperature inversion and the uncoupling with upper atmosphere when extremely stable boundary layers are formed. The cold bias of ARPEGE on the Antarctic Peninsula, especially in the winter, can largely be explained by atmospheric circulation errors, as these lead to an underestimation of mild and moist fluxes from the north-west towards the Peninsula.

The grounded AIS total precipitation in the ARP-AMIP simulation is extremely close to the estimates using the MAR or RACMO2 RCMs. However, the higher sublimation (and run-off) rates in the ARPEGE simulation compared to MAR and RACMO2 yields lower SMB values for the grounded AIS. Nevertheless, estimates of the AIS SMB using ARPEGE concurs independent estimates using satellites data (e.g., Vaughan et al., 1999; Arthern et al., 2006).

Many of the differences in the spatial distribution of precipitation rates between the ARP-AMIP simulation and MAR-ERA-I are linked to errors in atmospheric general circulations. These are for instance precipitation overestimates by ARPEGE over Marie-Byrd Land, the eastern part of the Peninsula and Dronning Maud Land, as well as precipitation underestimates over central West Antarctica and the west coast of the Peninsula.

## 4.2 Effects of Sea Surface Conditions

In the historical climate, we found that when driven by SSC from NorESM1-M, ARPEGE simulates significantly higher precipitation rates at the scale of the ice sheet (+218 Gt yr$^{-1}$, 2.2 $\sigma$). When driven by MIROC-ESM SSC, runoff and snow sublimation were found significantly lower due to cooler temperatures in spring and summer. In the following section, we
discuss the effects of SSC on simulated climate change, the consistency of the atmospheric model response between historical and future climate as well as the implication of SSC slection

### 4.2.1 Climate change signals

NorESM1-M and MIROC-ESM were chosen in this study because they display very different RCP8.5 projections in terms of changes in sea ice around Antarctica (respectively -14% and -45% of winter SIE) for late 21$^{st}$ century. The increase in SST
below 50 °S is much larger in MIROC-ESM (+ 1.8 K) than in NorESM1-M (+ 0.4 K). The separate effects of decreases in sea ice cover and increases in SST on Antarctic SMB has been assessed in Kittel et al. (2018) using the MAR RCM. Both result in an increase in Antarctic SMB (precipitation) that mostly takes places over coastal areas, as a result of increases in evaporation, saturation water vapour pressure, and decrease of the cover effect of sea ice. van Lipzig et al. (2002) found similar results using the RACMO RCM. In this study, we confirm the high impact of SSC on Antarctic SMB with a global atmospheric
model used at a high resolution similar to those commonly used ($\sim$ 30-50 kms) for Antarctic studies using RCMs. van Lipzig et al. (2002) have also investigated the separated effect of surface warming of the ocean and of homogeneous warming of the atmospheric column at the border of the domain of integration, the latter being more important as a result of increased moisture advection towards the ice sheet over a thicker atmospheric column. These two studies carried out with RCMs driven by climate reanalyses do not account for the response of the atmospheric general circulation to changes in oceanic surface conditions and
changes in radiative forcing as expected for the current century. This was done in Krinner et al. (2014) using LMDZ AGCM in a stretched-grid configuration who found that the effects of changes in oceanic surface conditions on Antarctic precipitation is much larger than the effect of changes in radiative forcings. As in Krinner et al. (2014), we found using an AGCM at a higher resolution that regional precipitation increases depend on the SSC source and on whether they are bias-corrected or not. It was also found in this previous study that the thermodynamic component (changes in precipitation for a given type of
atmospheric circulation patterns) was larger that the dynamic one (changes in precipitation due to changes in frequencies of a given atmospheric circulation pattern) in the projected increase in Antarctic precipitation.

In the projections presented in this study, the Antarctic increase in annual mean T$_{2m}$ and the relative increase in precipitation for late 21$^{st}$ century are within the range of the CMIP5 ensemble RCP8.5 (e.g., Palerme et al., 2017). Unsurprisingly, the warming obtained with projections using SSC from NorESM1-M (around +2.8K) belongs to the lower end of the values for
RCP8.5 CMIP5 projections, a consequence of weaker changes in the Southern Ocean SSC in this projection. In projections using MIROC-ESM SSC, the increase in annual T$_{2m}$ is around +4 K. The relative increase in precipitation in ARP-MIR-21-OC (+37%) belongs to the upper limit of the CMIP5 ensemble. As suggested by Krinner et al. (2010), the choice of the AOGCM providing SSC strongly influences the warming and precipitation increases obtained at the scale of the Antarctic continent.

Using NorESM1-M and original SSC from MIROC-ESM, the SMB (precipitation) increase obtained with ARPEGE ranges around 5.2 %.K$^{-1}$ (6.6 and 7.9 %.K$^{-1}$). This is within the range of values obtained in previous studies (Agosta et al., 2013; Ligtenberg et al., 2013; Krinner et al., 2014; Bracegirdle et al., 2015; Frieler et al., 2015; Palerme et al., 2017). Using bias-corrected SSC from MIROC-ESM, the sensitivity of the precipitation to temperature increase (8.8%.K$^{-1}$) is slightly above the higher end values of previous studies. Yet, this value is consistent with upper values of the CMIP5 ensemble (see Bracegirdle et al. (2015), Fig. 3) which mostly come from AOGCMs with large SIE in their historical simulations, and consequently larger decrease in sea ice in their future climate projections (Agosta et al., 2015; Bracegirdle et al., 2015). This suggests that there are some non-linearities in the sensitivity of Antarctic precipitation change to regional warming, as it is also sensitive to the reduced cover effect of sea ice. Consistent with findings from van Lipzig et al. (2002), we found that for regional warming within the + 3 to 4 K range, the increase in SMB is still largely dominated by precipitation increases, which remain much larger than the increase in surface melt and rain.

For the RCP8.5 simulation using SSC from NorESM1-M, the use of bias-corrected SSC has not yielded significantly different climate change signals with respect to the simulation using uncorrected SSC. For future projections with SSC from MIROC-ESM, using bias-corrected SSC led to significantly different climate change signals for many variables, especially in winter. In the projection with original MIROC-ESM SSC, the deepening of the low pressure zone around Antarctica is mainly organized in a three-wave pattern in JJA, while it shows a dipole in the projection with bias-corrected SCC. These differences lead to significantly different changes in atmospheric temperatures (0.8 K greater in ARP-MIR-21-OC in winter), the most dramatic difference being the larger (2 K) increase in west Marie-Byrd Land using bias-corrected SSC. Differences in atmospheric circulation are also unsurprisingly associated with significantly different changes in total precipitation. At the continental scale, the increase in moisture advection approximated trough P-E is 9% larger in ARP-MIR-21-OC than in ARP-MIR-21. The consequences of the three-wave pattern decrease in SLP around Antarctica in ARP-MIR-21 are obvious with three regions of lower precipitations increases with respect to ARP-MIR-21-OC. At the regional scale, it is noteworthy that all projections agree on a (slight) precipitation decrease in Marie-Byrd Land and the western Ross Ice Shelf (see Fig. 9). The decrease in precipitation in this region is however mitigated when using both set of bias corrected SSC. A lower increase or a slight precipitation decrease in Marie Byrd Land were also found in other studies (Krinner et al., 2008; Lenaerts et al., 2016). These results however bear uncertainties as many free AGCM (including ARPEGE) struggle to reproduce the depth and the variability of the Amundsen Sea Low. The changes in precipitation (and SMB) in this area are also extremely sensitive to the selected SSC. The changes in surface climate in the ASL area are extremely important for the SMB of the Antarctic Ice Sheet as a whole as glaciers of the Amundsen Sea Embayment are largely responsible for the positive contribution of the AIS to sea-level rise over recent years (e.g., Shepherd et al., 2018). The melting of ice shelves in this area is also expected to trigger the destabilization of glaciers located upstream (Rignot et al., 2013; Fürst et al., 2016; Deb et al., 2018).

Climate change signals for temperature and precipitation over large ice shelves (Ross and Ronne-Filchner) do not seem to substantially differ those from adjacent areas. Yet, as for the reconstruction of recent climate, projected climate change over these areas should be considered with caution, especially for near-surface temperatures.

### 4.2.2 Consistency of atmospheric model responses

The late winter (August to October, ASO) and late summer (February to April, FMA) errors of historical SST and SIC from NorESM1-M and MIROC-ESM with respect to observations are displayed in the supplementary material (Fig. A1 and A2). The same differences between SSC of their RCP8.5 projection and their bias-corrected equivalent are also shown. The differences in SSC used to drive the atmospheric model are, unsurprisingly, extremely similar between historical and future climate experiments.

Has the introduction of the same SSC "biases" with respect to the observed or bias-corrected references yielded the same responses of the atmospheric model in the historical and future climates? The consistency of the response of the atmospheric model is considered here as being the key for having the same climate change signals.

For simulations using SSC from the NorESM1-M model, the consistency of the response of the atmospheric model is clear. The similarities in the differences between ARP-NOR-20 and ARP-AMIP with differences between ARP-NOR-21 and ARP-NOR-21-OC is clear for many climate variables (SLP, see Fig. D2, 500 hPa geopotential, stratospheric temperatures, 500hPa zonal wind and near-surface atmospheric temperatures). In this perspective, the most interesting feature is that in both historical and future climate, the ARPEGE simulations forced by NorESM1-M original SSC are about 10% wetter at the Antarctic continental scale than their bias-corrected reference. The link here between the dynamical response of the atmospheric model and the SST biases of the NorESM1-M AOGCM seems physically consistent. NorESM1-M SSTs are indeed characterized by a warm bias in Southern hemisphere mid-latitudes (40-60°S) and a cool bias in the southern Tropics (see Fig. A3a), which cause a smaller meridional SST gradient. The response of the atmospheric model here is an increase in the moisture transport towards Antarctica (P-E larger by about 10%) and explains the additional $\sim 200$ Gt.yr$^{-1}$ ($2\,\sigma$) of precipitation on the ice sheet in the simulations realized with NorESM1-M non-corrected SSC. The consistency of the response of the atmospheric model in historical and future climate explains the absence of significant differences in the climate change signals between experiments with the original NorESM1-M SSC and their bias-corrected reference.

The consistency of the response of the atmospheric model is less clear for the projections realized with SSC from MIROC-ESM. Some changes in the differences between simulations forced with original SSC and those forced by their bias-corrected references are noticeable in winter and autumn SLP and zonal wind speed (an example for SLP can be seen in Fig. D3). The main result here, as a consequence of these differences, is a total precipitation difference in the RCP8.5 experiment with bias-corrected SSC of about +180 Gt yr$^{-1}$ ($\sim 1\sigma$), while there was almost no difference in total precipitation in the historical period between ARP-AMIP and ARP-MIR-20. Here, the link between biases in Southern Hemisphere SST from MIROC-ESM (see Fig. A3) and the response of ARPEGE appears less clear. SSTs from MIROC-ESM are mainly characterized by a cold bias in the Tropics throughout the years. With respect to the ARP-AMIP simulation, ARP-MIR-20 is also characterized by cooler temperatures throughout the tropical troposphere, much lower upper tropospheric and stratospheric temperatures in Antarctica. This suggests that interactions between SST biases, tropical convection, and stratospheric meridional temperature gradients could also explain the response of the atmospheric model when forced by MIROC-ESM SSC.

### 4.2.3 Implication of Sea Surface Conditions selection

In many cases, it has been reported that selecting the best skilled models for a given aspect of the climate system helps in better constraining the associated uncertainties on the climate change signal (e.g., Massonnet et al., 2012). Here, because we use bias-correction of the SSC, this aspect has reduced importance. While performing a limited number of climate projections, we cover a large range of the uncertainties associated with the evolution of the Southern Ocean surface condition for the Antarctic climate because it was shown to be its primary driver (Krinner et al., 2014). This approach is supported by the fact that biases of large-scale atmospheric circulation of coupled climate models were shown to be highly stationary under strong climate change (Krinner and Flanner, 2018), and that the response of the ARPEGE atmospheric model to the introduction of the same SSC "bias" was shown to be mostly unchanged in future climate.

The warming signal for the AIS in the CMIP5 model ensemble RCP8.5 projection is evaluated to be 4±1 K (Palerme et al., 2017). By selecting NorESM1-M and MIROC-ESM, we explored the range of the Southern Hemisphere SIE changes among the CMIP5 ensemble. However, using these SSC, the ARPEGE AGCM simulates a warming in the range of 2.8 to 4.2K, which is in the lower half of the range simulated by the CMIP5 models. Bracegirdle et al. (2015) found that about half of the variance of the CMIP5 projection in RCP8.5 scenario for Antarctic temperature and precipitation is explained by historical biases and sea ice decreases by the late 21 $^{st}$ century. Consequently, a non-negligible part of the uncertainties of Antarctic climate change is also linked to the representation of general circulation in the atmospheric model (Bracegirdle et al., 2013). This issue should be assessed in future work.

## 5 Summary and Conclusion

This study presented the first general evaluation of the capability of the AGCM ARPEGE to reproduce atmospheric general circulation of the high southern latitudes and the surface climate of the Antarctic Ice Sheet. ARPEGE is able to correctly represent the main features of atmospheric general circulation, although we have shown a negative bias in sea-level pressures at mid-latitudes and a positive bias around Antarctica especially in the Amundsen Sea sector. Unsurprisingly, the use of observed sea surface conditions (ARP-AMIP simulation) rather than SSC from NorESM1-M and MIROC-ESM helped to improve the representation of sea-level pressures in the southern latitudes in all seasons but summer. ARPEGE is also able to correctly reproduce surface climate of Antarctica except for large ice shelves. The differences in $T_{2m}$ with polar-orineted RCM MAR and *in-situ* observations is encouraging, especially given the large biases that are exhibited in other GCMs or even reanalyses when Antarctic surface climate is considered (Fréville et al., 2014; Bracegirdle and Marshall, 2012). Regarding precipitation, our estimates at the continental scale agree with estimates from other studies such as those using MAR or RACMO2, even though higher sublimation and run-off rates in ARPEGE yield smaller estimates of the GIS SMB by about 150 Gt yr$^{-1}$) (1.5 $\sigma$). Concerning regional patterns, the distribution of precipitation in the ARP-AMIP simulation differs from the one in the MAR RCM mainly as a consequence of errors in atmospheric general circulation.

The future climate projections presented in this study are among the first Antarctic climate projections realized at a "high" (Cordex-like) horizontal resolution using a global atmospheric climate model. Concerning climate change signals, we evaluate

the impact of using original and bias-corrected sea surface conditions from MIROC-ESM and NorESM1-M, which display opposite trends in their RCP8.5 projections for the Southern Ocean's late 21$^{st}$ century SIE (respect. -45% and -14% for winter SIE). Using SSC from NorESM1-M model, no significant differences in yearly or seasonal mean $T_{2m}$ increase, precipitation, or SMB changes were found when using bias-corrected SSC. When using SSC directly from MIROC-ESM model, the increase in precipitation is +30%, and it reaches +37% when using the corresponding bias-corrected SSC. This difference is statistically significant and is linked with clearly different dynamical and thermodynamical changes in SLP around Antarctica, occurring mainly in winter and spring. At the regional scale, large differences in $T_{2m}$ and precipitation increases are found when using bias-corrected SSC both from NorESM1-M and MIROC-ESM.

The analysis of the climate projections is further evidence the potential of the ARPEGE model for the study of Antarctic climate and climate change. When using SSC from NorESM1-M, we found a 10% higher precipitation accumulation at the continent scale (which is detrimental to the model skills for precipitation) with respect to the bias-corrected reference in both historical and future climate. These findings advocate once more for the use of bias-corrected SSC to drive climate projections using an AGCM. Additionally, this method reduces the uncertainty of the baseline (historical) climate and the need for computational resources as only one historical simulation using observed SSC in needed.

In this study, we confirm the importance of the coupled model choice from which SSC scenario is taken. By performing bias correction of SSC, we showed that not only the regional pattern of temperature and precipitation changes can be different but also the integrated changes in SMB and seasonal temperatures at the ice sheet scale. Unsurprisingly, projections using climate changes signal from MIROC-ESM SSC projections (larger decrease in sea ice) show higher increases in temperature and precipitation that the one using NorESM1-M SSC. This confirms the effect of sea ice decreases and SST increases on Antarctic temperatures and SMB in a "realistic" climate projection experiment. For the range of Antarctic warming obtained (+3 to +4 K), we confirm results from previous studies showing that the increase in SMB is largely dominated by increases in snowfall which remain much larger that the increase in melt and rainfall at the ice sheet scale. Considering changes in SIE at the two extreme end values from the CMIP5 ensemble, differences in Antarctic warming obtained ($\sim$ 1 K) are clearly smaller than the spread of CMIP5 projections for the AIS. This is consistent with the fact that a large part of the CMIP5 diversity for Antarctic climate projections comes from atmospheric model (errors) and associated uncertainties. Climate projections presented in this study still bear considerable uncertainties. These mostly come from ARPEGE errors (even when driven by observed SSC) on southern high latitudes general atmospheric circulation, which casts some doubt on the reliability of the projected Southern Hemisphere atmospheric circulation changes. As a consequence, in future work, we will assess the impact of AGCM atmospheric circulation errors by performing an ARPEGE simulation nudged towards the reanalysis and use the statistics of the model drift in this nudged simulation such as done in Guldberg et al. (2005) to perform an atmosphere bias-corrected ARPEGE historical simulation. Bias-corrected projections such as done in Krinner et al. (2019) can then also be assessed using the method presented in this study.

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

which is responsible for CMIP, and we thank the climate modeling groups participating to CMIP5 for producing and making available their model output. For CMIP the U.S. Department of Energy's Program for Climate Model Diagnosis and Intercomparison provides coordinating support and led development of software infrastructure in partnership with the Global Organization for Earth System Science Portals.

The Centre National de Recherches M'et'eorologique (M'et'e-France, CNRS) and associated colleagues are warm-fully thanks for providing resources and help to run ARPEGE model.

We also thank the Scientific Committee on Antarctic Research, SCAR and the British Antarctic Survey for the availability of the MET READER data base. We also want to thanks warmfully Michiel van den Broeke for providing access to the latest ERA-Interim driven RACMO2 outputs for Antarctica. We thanks Jan Lenaerts and anonymous referee for reviews and comments aiming at the improvement of the manuscript.

## Appendix A: Sea Surface Conditions

In this section, we present the historical bias in SSC in MIROC-ESM and NorESM1-M (Fig. A1) used to force ARPEGE model as well as the differences between SSC in rcp8.5 scenarios in these model and their bias-correction (Fig. A2). The skills of the bias-correction method for SSC can be appreciated as the similarity between differences in futures SST is striking. For SIC, the pattern of the model bias in historical climates can easily be identified in the differences between original and bias-corrected SSC (Fig. A2), but because there is a decrease of SIE, these patterns are shifted poleward. Yearly and seasonal

**Table A1.** Annual and seasonal Southern Hemisphere mean historical Sea Ice Extent (SIE, $10^6$ km$^2$) in observations, NorESM1-M and MIROC-ESM.

|  | Year | DJF | MAM | JJA | SON |
|---|---|---|---|---|---|
| Observations | 9.6 | 4.4 | 5.6 | 13.5 | 14.7 |
| NorESM1-M | 9.8 | 4.8 | 6.6 | 14.0 | 15.4 |
| MIROC-ESM | 8.9 | 3.1 | 4.0 | 13.3 | 15.3 |

**Table A2.** Annual and seasonal Southern Hemisphere mean projected Sea Ice Extent and absolute change with respect to historical climate ($10^6$ km$^2$) in NorESM-1M and MIROC-ESM RCP8.5 projection and corresponding bias-corrected SSC.

|  | Year | DJF | MAM | JJA | SON |
|---|---|---|---|---|---|
| NorESM1-M-rcp85 | 8.2 | 4.0 | 5.1 | 11.7 | 13.6 |
| *Change ($10^6$ km$^2$)* | *-1.6* | *-0.8* | *-1.5* | *-2.3* | *-1.8* |
| NorESM1-M-rcp85-bc | 7.9 | 3.5 | 4.2 | 11.1 | 12.7 |
| *Change ($10^6$ km$^2$)* | *-1.6* | *-0.8* | *-1.5* | *-2.3* | *-1.8* |
| MIROC-ESM-rcp85 | 4.2 | 0.9 | 1.2 | 6.8 | 8.2 |
| *Change ($10^6$ km$^2$)* | *-4.7* | *-2.2* | *-2.8* | *-6.5* | *-7.2* |
| MIROC-ESM-rcp85-bc | 4.2 | 1.0 | 1.5 | 6.8 | 7.6 |
| *Change ($10^6$ km$^2$)* | *-5.3* | *-3.4* | *-4.1* | *-6.7* | *-7.1* |

South Hemisphere SIE in MIROC-ESM, NorESM1-M and observations (Table A1) and in the two AOGCM original and bias-corrected RCP8.5 projection (Table A2) are also presented in this supplementary material. Here again, the efficiency of the bias-correction methods to reproduce the climate change signal in hemispheric SIE from the coupled model is confirmed. In Figure A3, SST historical bias for both coupled models for each season in the southern hemisphere are displayed in order to support the discussion on how the atmospheric model has responded to the same SST biases or perturbations in present and future climate.

In Table A2, the climate change signals in SIE in scenarios from MIROC-ESM and NorESM1-M can be evaluated, with the decrease in sea ice being three times importanter in MIROC-ESM projection. It can also be noted that both AOGCM hemispheric SIE are relatively close to the observations. Only an underestimate of about 20% in summer and autumn SIE in MIROC-ESM can be mentioned.

## Appendix B: Near-surface temperature

In this section, we present additional material for near-surface temperatures (T$_{2m}$). The difference between T$_{2m}$ from the ARP-AMIP simulation and those from the MET READER data base and corresponding evaluation statistics can be seen in Table B1. The location of the weather station can be seen on the right panel of Fig. 4.

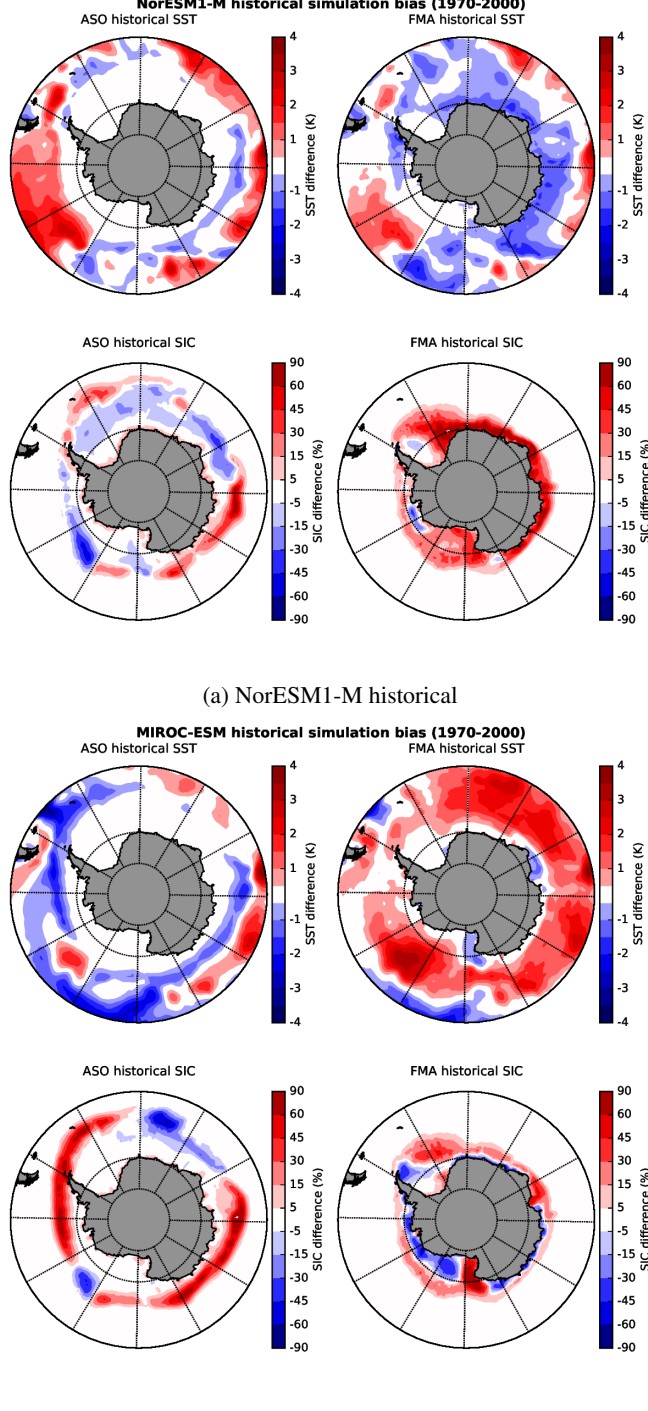

(a) NorESM1-M historical

(b) MIROC-ESM historical

**Figure A1.** Bias in SST (*top*) and SIC (*bottom*) for late winter, August, September, October(*left*) and summer, February, March, April (*right*) historical simulations of (*a*) NorESM1-M and (*b*) MIROC-ESM.

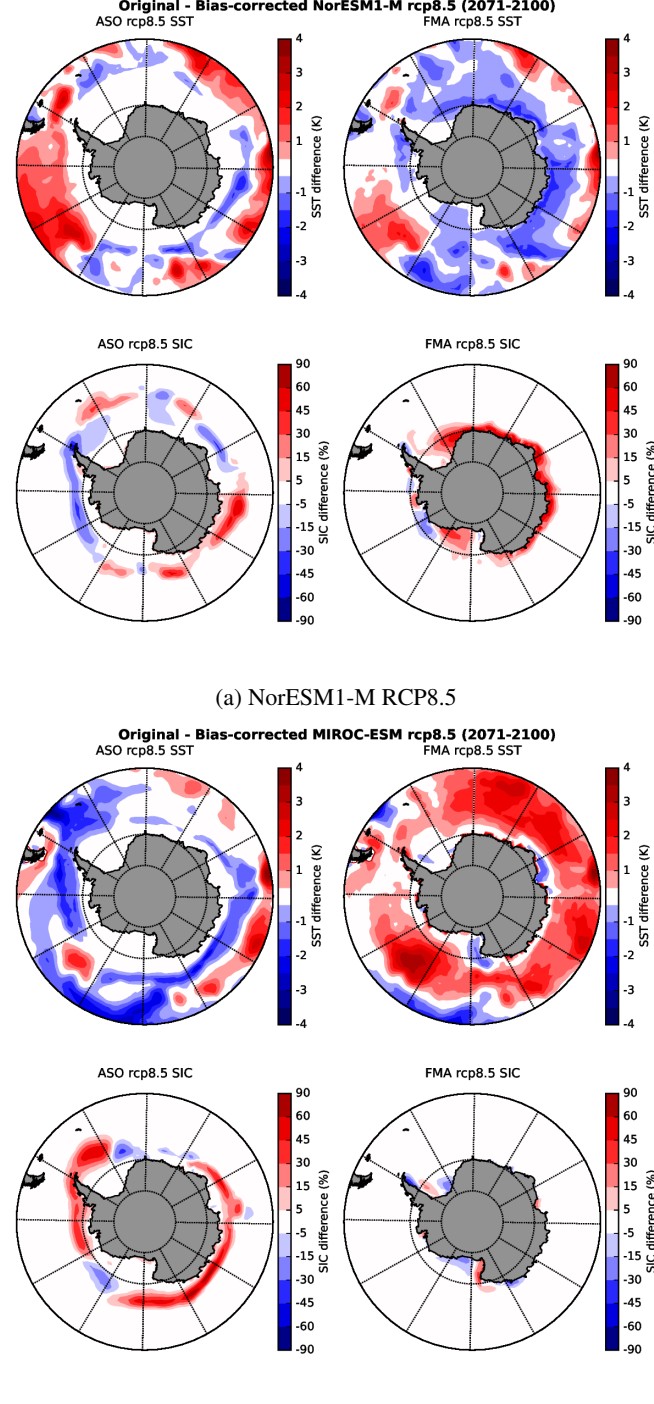

(a) NorESM1-M RCP8.5

(b) MIROC-ESM RCP8.5

**Figure A2.** Same as Fig.A1 but for RCP8.5 scenario and corresponding bias corrected SSC

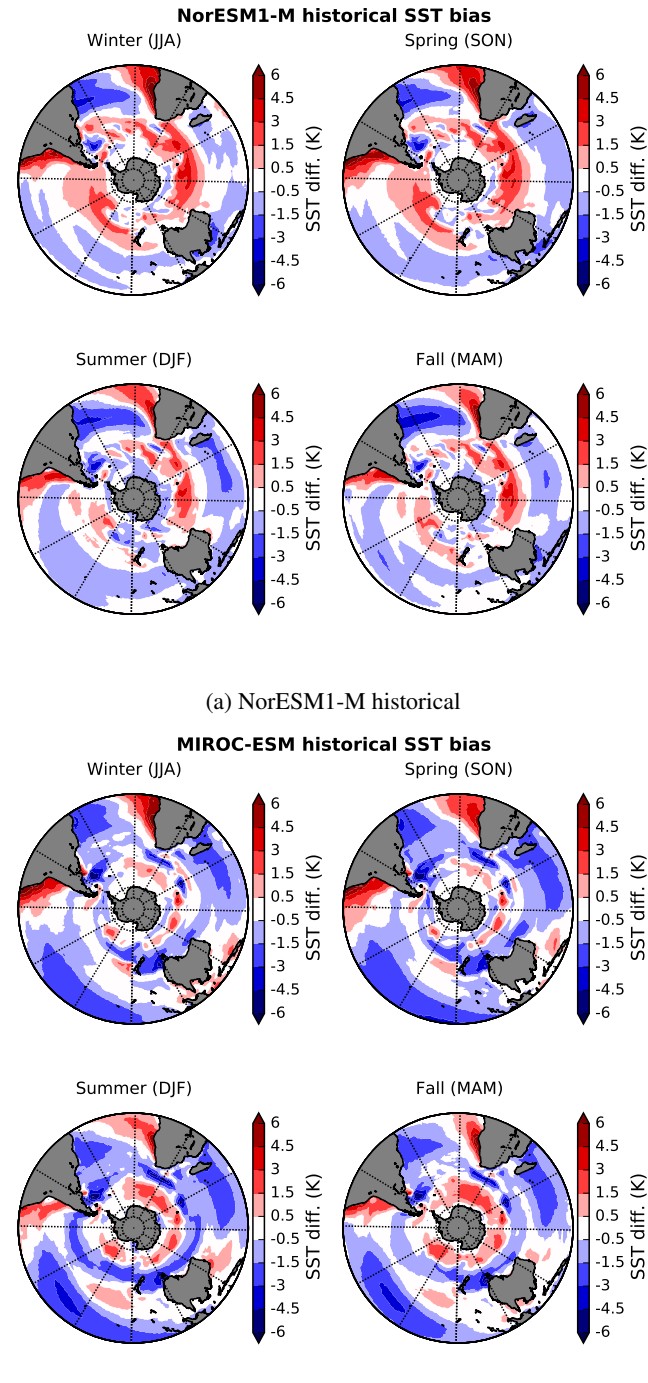

(a) NorESM1-M historical

(b) MIROC-ESM historical

**Figure A3.** Seasonal historical bias in SST in the Southern hemisphere from NorESM1-M (*top*) and MIROC-ESM (*bottom*).

The effect of introducing biased SSC on the modelling of Antarctic $T_{2m}$ with ARPEGE AGCM is also presented in Fig. B1. For ARP-NOR-20 (Fig. B1a), the introduction of biased SSC increase the warm bias on the East Antarctic Plateau (EAP) with respect to MAR and weather stations already present in ARP-AMIP (Fig. 4). The same statement can be made for the winter cold bias over the Peninsula. In summer, there are relatively few differences in the skills of the latter two simulations, which is consistent with similar errors on large-scale atmospheric circulation (Fig. 2).

For ARP-MIR-20 (Fig. B1a), the cold bias over the Peninsula is also larger than ARP-AMIP for both seasons. The winter warm bias over the EAP is similar than in ARP-AMIP. In summer, the general tendency of ARP-MIR-20 to be cooler than ARP-AMIP over the continent leads to a decrease of the warm bias with respect to MAR over the margins of the EAIS and WAIS on one hand, but increase the cold bias on the EAP on the other hand, which can be seen in the differences with MAR and weather stations.

## B1   Ice Shelves

In this section, we further investigate the causes of the large discrepancies between ARPEGE and MAR over ice shelves and try to evaluate which part these discrepancies are actually due to the systematic biases of each model. Over the large ice shelves (Ronne-Filchner and Ross) the ARP-AMIP simulation is systematically 7 to 10 K (up to 12 K over the center of Ross) warmer than MAR in winter, while in summer, it is 5-7 K cooler (Fig. 4). While no *in-situ* temperature records long enough to evaluate a freely evolving climate model such as ARPEGE is currently available for these areas, the MAR-ERA-I simulation has been evaluated against automatic weather station from the READER data base (Agosta, 2018). Over the Ross Ice Shelf, MAR shows an average systematic bias of -2.8 K with biases larger than 5 K for the coolest stations (center of the ice shelf). This suggests that about 1/3 of the MAR-ARPEGE discrepancy over large ice shelves in winter seems to actually comes from a MAR cold bias over these areas. This can also be seen over smaller ice shelves of the Dronning Maud Land area where ARPEGE is 5-7 K warmer in winter when compared to MAR, while ARPEGE biases with respect to Halley and Neumayer weather station located over ice shelves of this area are respectively only + 1.2 and + 0.9 K (Table B1). The evaluation in Agosta (2018) shows that MAR also has a $\sim$ 3 K cold bias over ice shelves in summer, which suggests that ARPEGE cold bias might be even larger during this season. This analysis seems to be confirmed in the comparison between ARP-AMIP and RACMO2 (Fig. B7) where ARPEGE "warm bias" over ice shelves is reduced over most of the Ross Ice Shelf ($< 5$ K) and almost completely disappears over Ronne-Filchner while ARPEGE "cold bias" over these areas in summer is more striking.

In the following, we examine differences between MAR-ERA-I and ARP-AMIP for different components of the surface energy balance (latent heat flux, sensible heat flux, downward long-wave radiation), albedo and near-surface temperature inversion. Unlike what has been done for near-surface temperature, wind speed, surface pressure and SMB, the MAR-ERA-I simulation has not been rigorously evaluated against observational data sets for these variables. As a consequence, here more than anywhere else, these comparisons are meant to help in understanding model-model differences rather than being an indirect evaluation of ARPEGE model.

In the version of ARPEGE used, ice shelves were not considered as land in the land surface model. To solve this issue, we forced the sea-ice concentration to be 100% and the sea ice thickness to be 40 meters in order to simulate realistic heat fluxes

**Table B1.** Error on READER weather station $T_{2m}$ (in K) in the ARP-AMIP simulation for the reference period 1981-2010. Errors significant at p=0.05 are presented in **bold**.

| Stations | DJF | MAM | JJA | SON |
|---|---|---|---|---|
| **EAP** | | | | |
| Amundsen Scott | 0.5 | **2.4** | 1.1 | 0.9 |
| Vostok | **-1.5** | **3.2** | **3.2** | **1.9** |
| *Mean error* | -0.5 | 2.8 | 2.1 | 1.4 |
| *RMSE* | 1.1 | 2.8 | 2.4 | 1.5 |
| **Coastal EA** | | | | |
| Casey | **-4.0** | **-5.7** | **-6.9** | **-5.4** |
| Davis | **-1.6** | **-4.2** | **-6.0** | **-3.3** |
| Dumont Durville | -0.5 | **-2.8** | **-4.1** | **-2.2** |
| Mawson | **-2.2** | **-4.3** | **-5.7** | **4.3** |
| McMurdo | **-7.1** | **-6.5** | **-8.1** | **-8.4** |
| Mirny | **-1.2** | **-2.2** | **-3.0** | **-2.0** |
| Novolazarevskaya | **2.5** | 0.6 | -1.0 | 0.6 |
| Scott Base | **-5.0** | **-3.1** | **-4.6** | **-5.0** |
| Syowa | -0.2 | -0.6 | **-1.5** | 0.0 |
| *Mean error* | -2.2 | -3.3 | -4.5 | -3.3 |
| *RMSE* | 3.5 | 3.9 | 5.1 | 4.3 |
| **Ice shelves** | | | | |
| Halley | **1.3** | **2.5** | 1.2 | 0.9 |
| Neumayer | **2.2** | **1.2** | 0.9 | **1.4** |
| *Mean error* | 1.7 | 1.8 | 1.1 | 1.2 |
| *RMSE* | 1.8 | 1.9 | 1.1 | 1.2 |
| **Peninsula** | | | | |
| Bellingshausen | **-1.0** | -0.4 | -0.2 | -0.1 |
| Esperanza | **-1.1** | 0.5 | -1.3 | -0.9 |
| Faraday | **-2.7** | **-4.7** | **-5.7** | **-3.7** |
| Marambio | **-1.9** | 1.0 | -1.3 | -1.6 |
| Marsh | **-0.8** | -0.4 | -0.3 | -0.0 |
| Orcadas | **-1.1** | -0.0 | 0.6 | -0.8 |
| Rothera | **-5.6** | **-7.9** | **-8.7** | **-6.1** |
| *Mean error* | -2.0 | -1.7 | -2.4 | -1.9 |
| *RMSE* | 2.6 | 3.5 | 4.0 | 2.8 |
| **Southern Ocean** | | | | |
| Gough | **-1.0** | -0.3 | 0.0 | **-0.8** |
| Macquarie | **-0.7** | -0.4 | 0.2 | **-0.5** |
| Marion | **-1.2** | **-0.4** | -0.1 | **-0.7** |
| *Mean error* | -1.0 | -0.4 | 0.0 | -0.6 |
| *RMSE* | 1.0 | 0.4 | 0.1 | 0.7 |

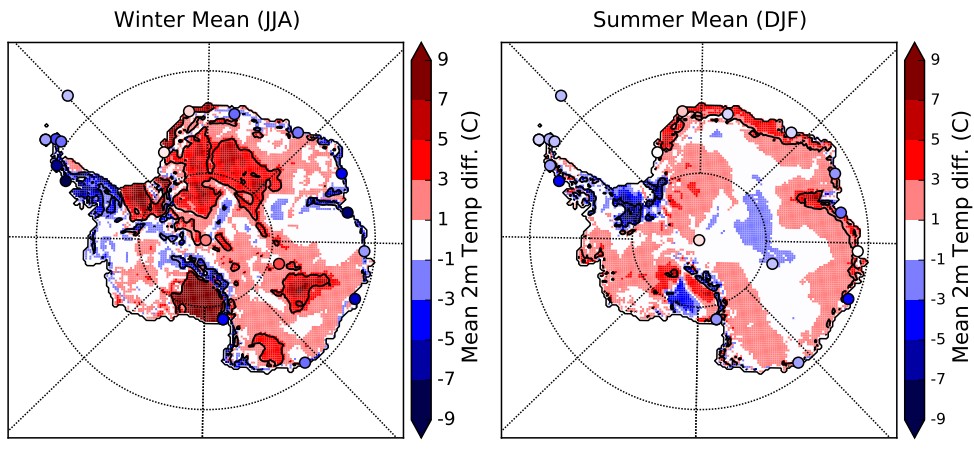

(a) ARP-NOR-20

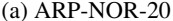

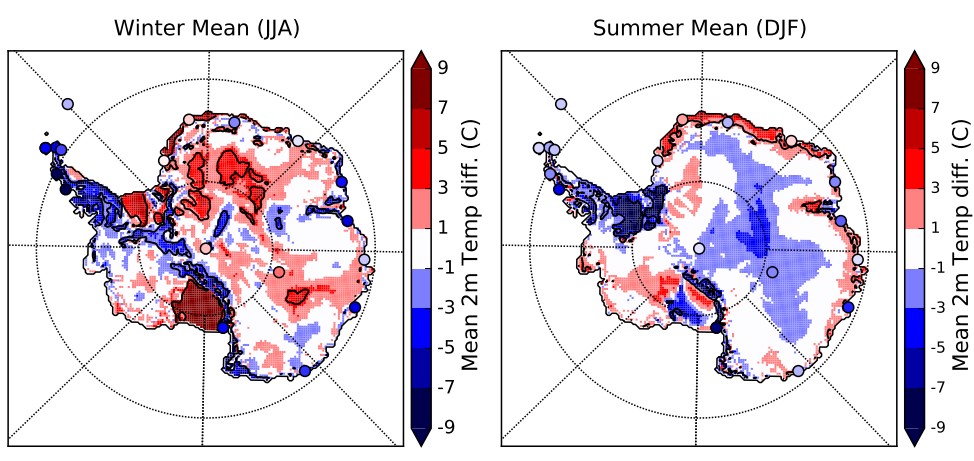

(b) ARP-MIR-20

**Figure B1.** $T_{2m}$ differences between ARP-NOR-20 (*top*) and ARP-MIR-20 (*bottom*) and MAR-ERA-I simulations in winter (JJA, *left*) and summer (DJF, *right*) for the reference period 1981-2010. Circles are $T_{2m}$ differences between ARP-AMIP and weather stations from the READER data base. Black contour lines represent areas where $\mid ARPEGE - MAR \mid > 1.MAR\sigma$.

at the surface. These modifications allowed to completely shut down latent heat fluxes from the surface (Fig. B2) and to have negative sensible heat fluxes (heat transfer from the atmosphere to surface, Fig.B3) in winter as expected, and in agreement with the fluxes modelled in MAR simulation. Thanks to the accumulation of snow on top of sea ice accounted for in GELATO, the effective albedo (SWU/SWD, Fig. B6) over ice shelves in ARPEGE compares reasonably well with MAR. This statement

is also valid for most of the ice sheet. The structure of the near-surface inversion has been investigated as another possible explanation for discrepancies between MAR and ARPEGE. To do so, we represent the difference between surface temperature ($T_s$) and the temperatures at 20 metres ($T_{20m}$) in both model and the corresponding difference (Fig. B6). Over large ice shelves, the seasonality of the differences (weaker near-surface inversion in ARPEGE in winter, and larger in summer) is consistent with the differences in near-surface temperatures between the two model along the seasons. This statement is also valid for the

very top of the high Antarctic Plateau where ARPEGE tends to be too warm (with respect to MAR and observations) in winter and slightly too cold in summer. This suggests that ARPEGE underestimates the strength of near-surface temperature inversion due to the formation of very stable boundary layer in winter as many climate models do (King et al., 2001; Bazile et al., 2014, e.g.,). Another part of the explanation for warmer ARPEGE temperatures over ice shelves in winter might also comes from higher latent and sensible fluxes over the sea ice area (see Fig. B3 and B2), which favours advection of warmer and moist air

over ice shelves. The cloudiness (*not shown*) and the downward longwave radiation (Fig. B4) over ice shelves being indeed higher in ARPEGE than in MAR.

Discrepancies between models for near-surface temperatures over large ice shelves and errors with respect to sparse *in-situ* observations even for polar-oriented RCMs widely used as reference (MAR and RACMO2) shows that there is still room for improvement and that these areas might be an even more challenging test cases for surface boundary layer scheme than the

high Antarctic Plateau.

## Appendix C:  Surface Mass Balance

### C1    Precipitation : comparison with MAR RCM

In this section, the effects of driving ARPEGE with biased SSC (NorESM1-M an MIROC-ESM) on the modelling of Antarctic precipitation are presented trough comparisons with MAR-ERA-I total precipitation. Differences between ARP-AMIP, ARP-

NOR-20 and ARP-MIR-20 with MAR-ERA-I for total precipitation are show in Fig. C1. Mean error and RMSE with respect to MAR are presented in the upper-left corner. The pattern of the errors is quite similar for each simulation. Unsurprisingly, the best agreement (smaller RMSE) with MAR is found the ARP-AMIP simulation. The wet biases with respect to MAR over Dronning Maud and Marie-Byrd Land also evidenced in ARP-AMIP tend to increase in both ARP-NOR-20 and ARP-MIR-20 simulations. The ARP-NOR-20 simulation has systematic wet bias (larger mean error) with respect to MAR at the continent

scale consistent with the 10% increase in precipitation integrated over the whole ice sheet found in this simulation with respect to ARP-AMIP.

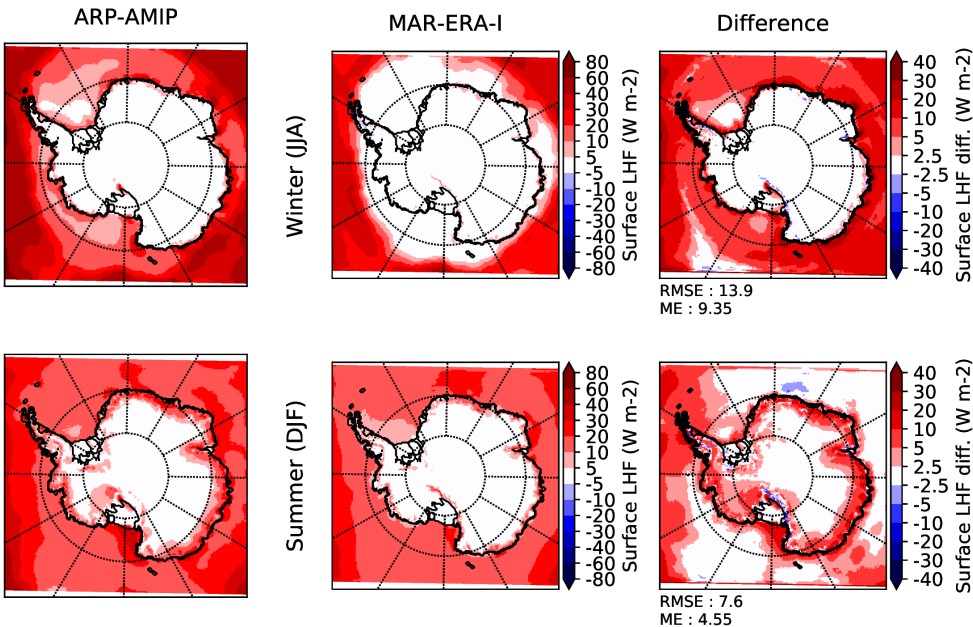

**Figure B2.** Mean surface latent heat flux (W m$^{-2}$) in ARP-AMIP (left), MAR-ERA-I (centre) and differences between the two models (right). The 1981-2010 mean flux over winter month (JJA) are shown on the upper part of the figure, while it is shown on the lower part for summer months (DJF).

## C2   Snowfall : comparisons with CloudSAT data set

Here, we present the comparisons of ARPEGE snowfall rates with those from the CloudSAT data set (Palerme et al., 2014, 2017). These results are not presented in the main part of the manuscript as they have to be considered with any caution for two main reasons. First, the CloudSAT data set is available for a very short period (2007-2010, and only north of 82°S) which generates many uncertainties when used to evaluate a thirty years series of precipitation coming from a freely evolving climate model. Second, CloudSAT snowfall rates are representative for snowfall rates 1200 m above the surface as these satellites measurements are too sensitive to ground clutter below this elevation (Palerme et al., 2014). Snowfall rates at the surface and 1200 m higher up can differ significantly near the ice sheet margins where the sublimation of falling snow particles within the dry katabatic layer present near the surface can be important (Grazioli et al., 2017) or over the center of the ice sheet where clear sky precipitation forming in the lowest layers of the atmosphere represent a substantial share of yearly precipitation (Palerme et al., 2014). Snowfall rates at different elevation were not kept as default diagnostic variables when running the ARPEGE simulations presented in this study and therefore snowfall rates from CloudSAT can only be compared with surface snowfall rates from ARPEGE such as done in (Palerme et al., 2017) for the comparisons with ERA-Interim reanalyses and the CMIP5 ensemble. Nevertheless, the comparison with the CloudSAT data set offers a unique opportunity to evaluate ARPEGE snowfall

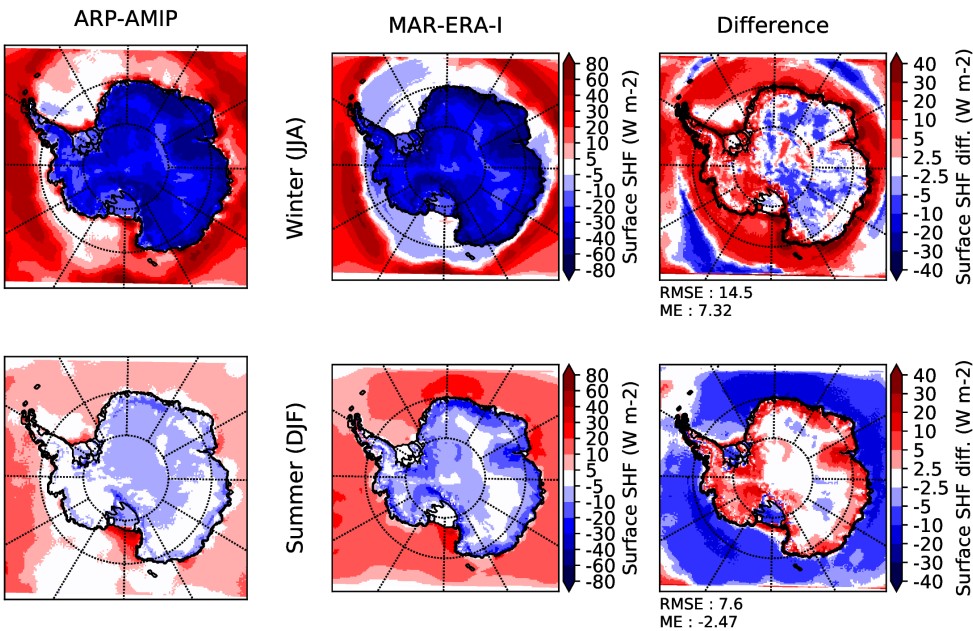

**Figure B3.** Mean surface sensible heat flux (W m$^{-2}$) in ARP-AMIP (left), MAR-ERA-I (centre) and differences between the two models (right). The 1981-2010 mean flux over winter month (JJA) are shown on the upper part of the figure, while it is shown on the lower part for summer months (DJF).

with a reliable and model independent precipitation data set in Antartica over a wide spatial coverage.

The difference with and ratio on CloudSAT snowfall rate for the three historical ARPEGE simulations presented in this study are shown in Fig C2. The snowfall rate averaged over the whole continent, the interior ($> 2250$ m a.s.l) and the peripheral ($<$ 2250 m a.s.l) AIS are reported in Table C1. In this table, we also reproduced the values from ERA-I and the CMIP5 ensemble found by Palerme et al. (2017). If we consider the ARP-AMIP simulation, the overestimation of the mean snowfall rate is about 30% of the CloudSAT value. ARPEGE estimate belongs to the lower half of the CMIP5 ensemble. Over the peripheral ice sheet ($< 2250$ m), the overestimation in ARPEGE is about 20% of CloudSAT value. For the high interior of the AIS ($> 2250$ m), the mean snowfall rate in ARPEGE is almost the double of what it is in CloudSAT while ARPEGE values are close to the minimum of the CMIP5 ensemble. However, the fact that CloudSAT measurements do not account for precipitation forming below 1200 m above the surface suggests that snowfall in this area are less reliable and most likely underestimated (Palerme et al., 2014). The agreement between ARPEGE and ERA-Interim is better in this area with overestimation in ARPEGE still reaching about 50%.

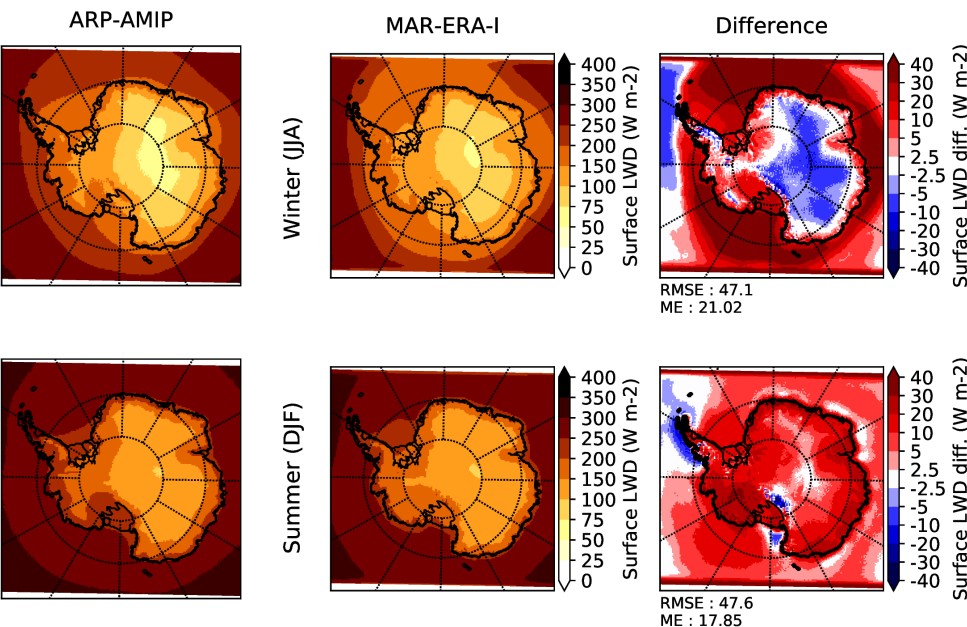

**Figure B4.** Mean surface longwave downward radiation (W m$^{-2}$) in ARP-AMIP (left), MAR-ERA-I (centre) and differences between the two models (right). The 1981-2010 mean flux over winter month (JJA) are shown on the upper part of the figure, while it is shown on the lower part for summer months (DJF).

**Table C1.** Yearly mean snowfall rate (mm.we.yr$^{-1}$) integrated north of 82°S over the whole continent, the interior of the AIS ($> 2250$ m) and the peripheral ice sheet ($< 2250$ m) Values for ARPEGE simulations are for 1981-2010 after interpolating ARPEGE snowfall on ClouSAT 1 grid. [1] CloudSAT values for 2007-2010 from Palerme et al. (2017), values in parenthesis are the found using ARPEGE land-sea mask. [2] ERA-Interim, CMIP5 ensemble minimum, average and maximum values from Palerme et al. (2017) over 1986-2005.

| | Continent | $> 2250$ m | $< 2250$ m |
|---|---|---|---|
| ARP-AMIP | 213 | 60 | 335 |
| ARP-NOR-20 | 230 | 69 | 360 |
| ARP-MIR-20 | 210 | 60 | 331 |
| CloudSAT[1] | 172 (165) | 36 (29) | 306 (271) |
| ERA-I[2] | 165 | 46 | 279 |
| CMIP5$_{min}$[2] | 158 | 50 | 254 |
| CMIP5$_{avg}$[2] | 224 | 74 | 363 |
| CMIP5$_{max}$[2] | 354 | 110 | 611 |

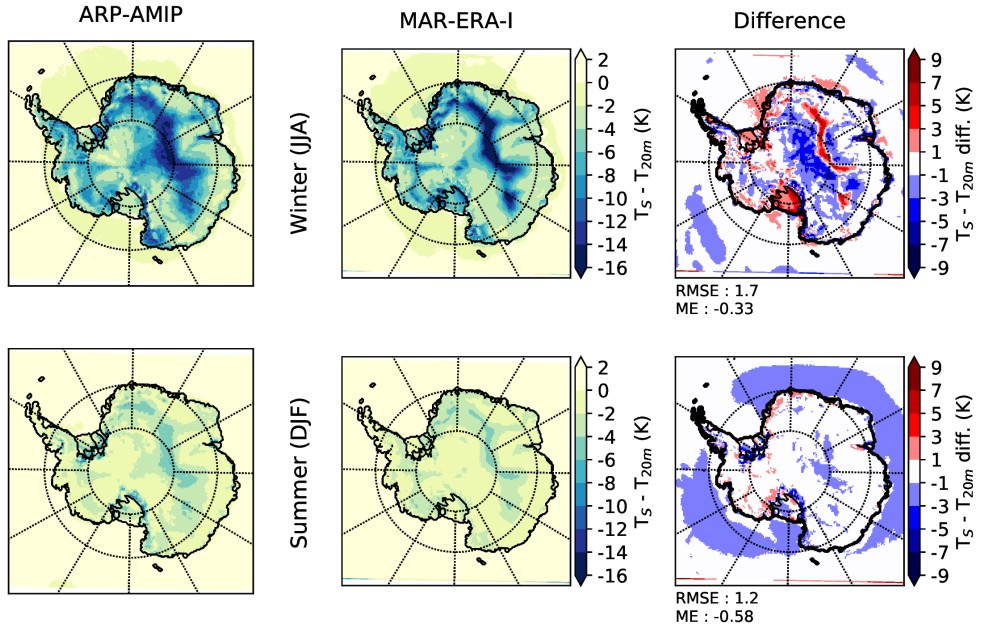

**Figure B5.** Mean near-surface temperature inversion ($T_S$ - $T_{20m}$, in K) in ARP-AMIP (left), MAR-ERA-I (centre) and differences between the two models (right). The 1981-2010 mean for winter month (JJA) are shown on the upper part of the figure, while it is shown on the lower part for summer months (DJF).

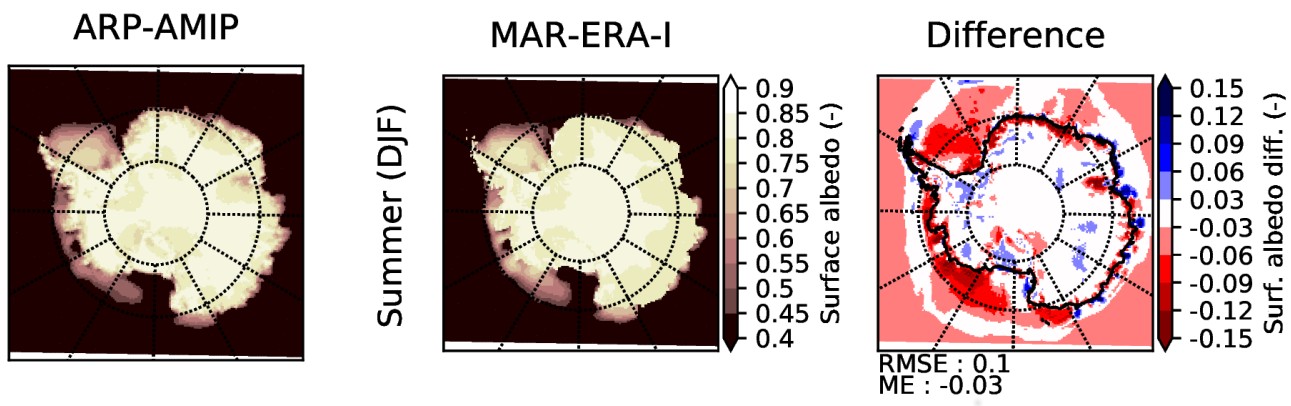

**Figure B6.** Mean surface summer (DJF) effective albedo ($SWU/SWD$) in ARP-AMIP (left), MAR-ERA-I (centre) and differences between the two models (right).

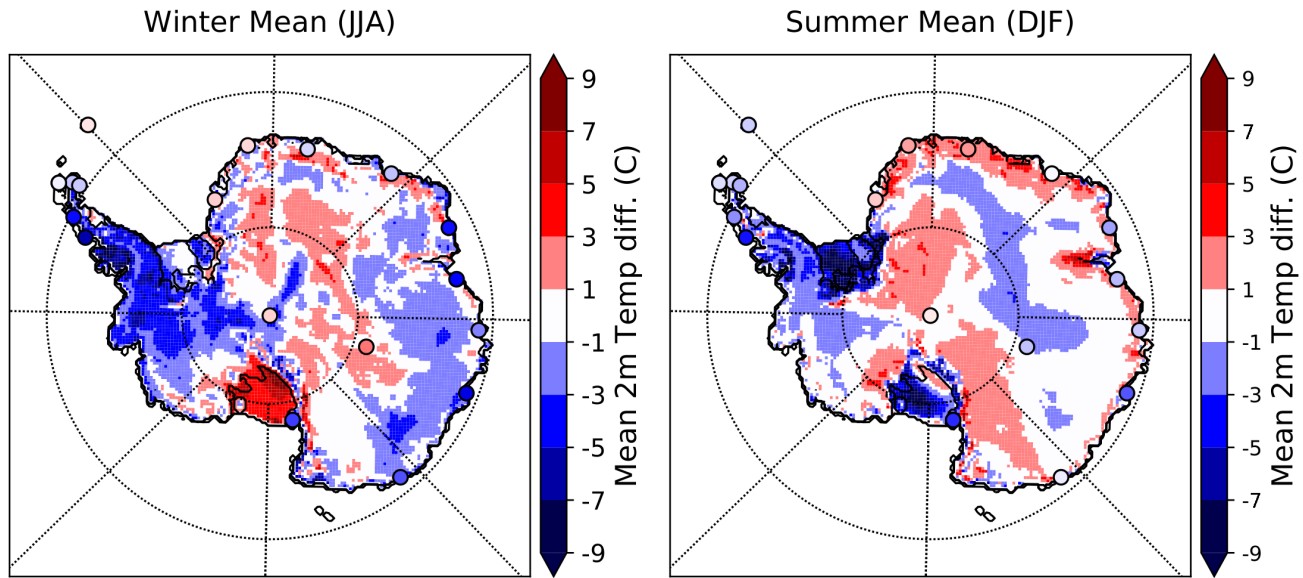

**Figure B7.** T$_{2m}$ differences between ARP-AMIP and ERA-I driven RACMO2 (van Wessem et al., 2018) in winter (JJA, *left*) and summer (DJF, *right*) for the reference period 1981-2010. Circles are T$_{2m}$ differences between ARP-AMIP and weather stations from the READER data base.

## C3 Surface melt

In this section, we present and briefly discuss additional results from the comparisons between ARPEGE and polar-oriented RCMs MAR and RACMO2. It can be seen in Table 4 that compared to reference RCMs MAR and RACMO2 driven by ERA-I reanalyses, ARPEGE represents reasonably the total integrated melt flux at the surface of the grounded AIS as the yearly mean
in ARP-AMIP falls within the $\pm 1.\sigma$ of the estimation using RACMO2 (Agosta et al., 2018; van Wessem et al., 2018) while the difference with MAR is +1.9 $\sigma$ of MAR standard deviation. In Fig. C3 and Fig. C4, one can see that the spatial distribution of melt areas over the AIS is reasonably represented in ARP-AMIP simulation if MAR and RACMO2 are taken as reference. In comparison with both RCMs, some limitation of ARPEGE model can however be mentioned : i) an underestimation of melt intensities over coastal areas and small ice shelves on the west and east side of the AP, consistent with ARPEGE errors on
atmospheric general circulation and identified cold biases over these areas due underestimated warm and moist air advection from the north-west and possibly reduced Foëhn event frequencies on the east side of the Peninsula (Larsen Ice Shelf) ii) overestimated melt intensities over the ridge of the narrow northern part of the Peninsula likely due to poorer representation of the topography due to coarser ARPEGE horizontal resolution over this area ($\sim$ 45 kms vs 35 kms in MAR and 27 kms in RACMO2) iii) overestimation of melt intensities over large ice shelves (Ronne-Filchner and Ross) consistent with reduced
ARPEGE skills for the representation of surface boundary layer processes over these areas. Despite these limitations, it can be

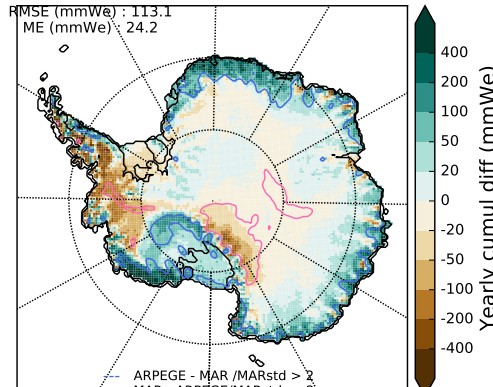

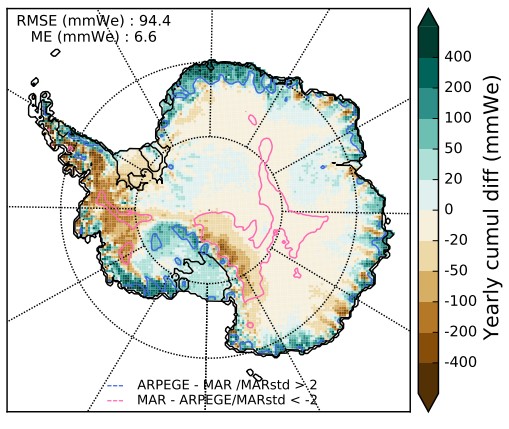

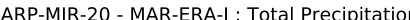

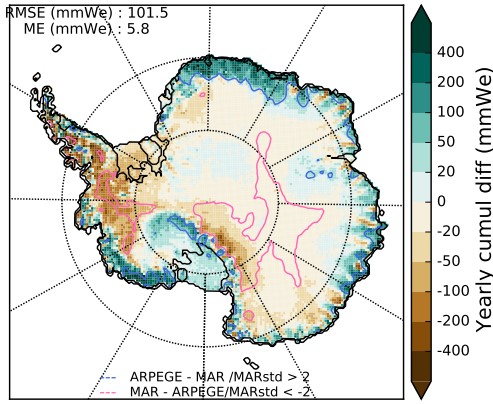

**Figure C1.** ARP-AMIP(*top*), ARP-NOR-20(*centre*) and ARP-MIR-20(*bottom*) minus MAR-ERA-I total precipitation. Pink and blue contour lines indicates where difference is larger than two MAR standard deviation (2-$\sigma$). RMSE and mean error with respect to MAR are indicated in the upper-left corner.

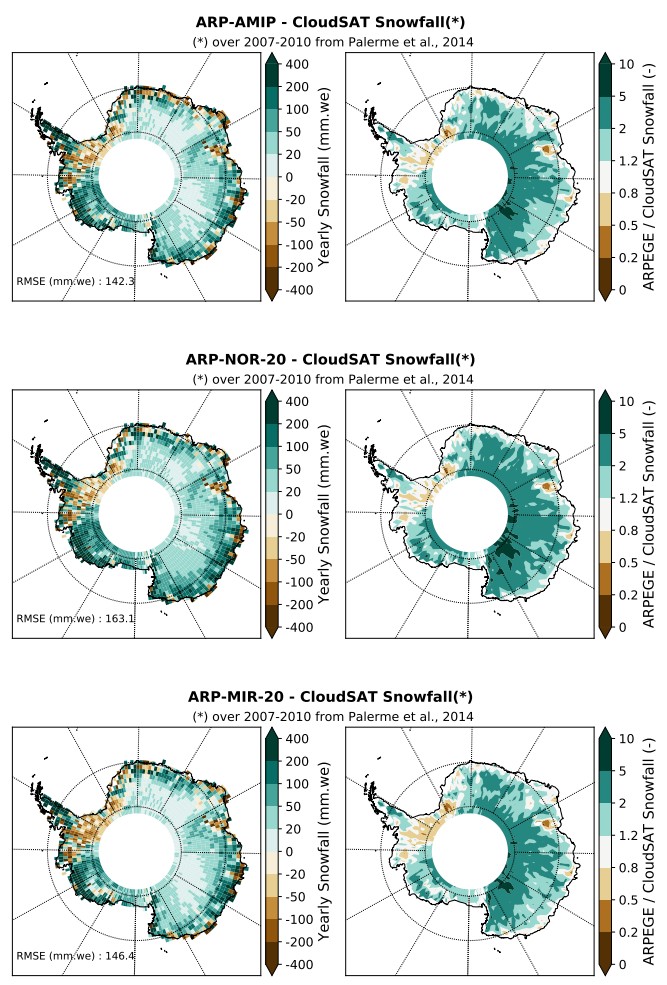

**Figure C2.** Left pannel : ARP-AMIP(*top*), ARP-NOR-20(*centre*) and ARP-MIR-20(*bottom*) over 1981-2010 minus CloudSAT (2007-2010) yearly mean snowfall (mm.we.yr$^{-1}$). Right pannel : ratio of ARPEGE on CloudSAT yearly mean snowfall.

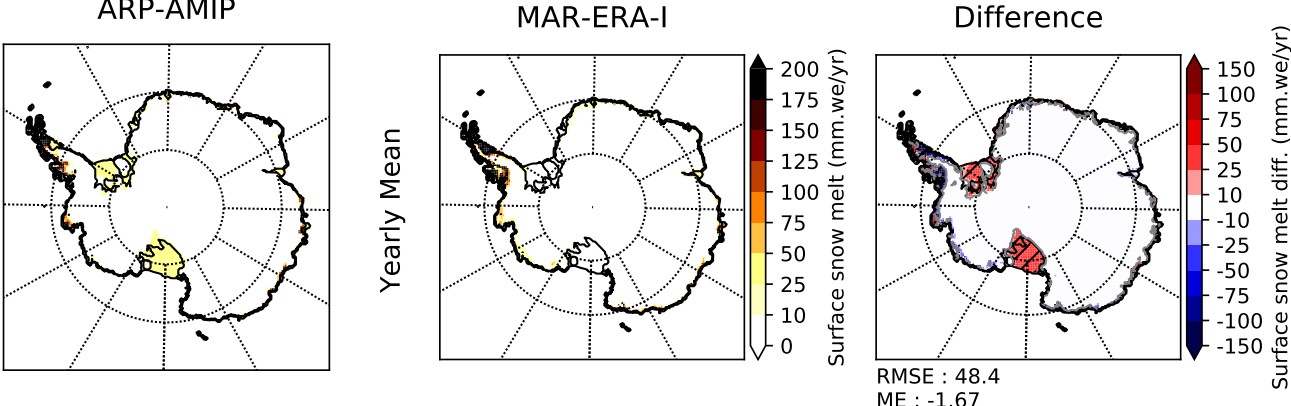

**Figure C3.** Yearly mean surface snowmelt (mm.we yr$^{-1}$ ) in ARP-AMIP (left), MAR-ERA-I (centre) and differences between the two models (right). Grey-contoured, hashed areas indicate where the difference is larger than 1 MAR standard deviation.

assumed that ARPEGE represents reasonably surface melt fluxes over the grounded AIS. This statement is however no longer valid if we consider surface run-off, as about $1/3$ of surface liquid water inputs leaves the snowpack in ARPEGE simulations (see Table 4), while this fraction is only 1 to 2 % in MAR and RACMO2. This shows some limitations of ISBA-ES snow scheme for the representation of the retention capacity of the Antarctic snow pack. As a result, projected changes in surface
5  run-off are not presented or discussed in section 3.2 due to limited ARPEGE skills for this variable in present climate and because of strong non-linearities often observed in changes in surface run-off in a warming climate.

## Appendix D: Atmospheric general circulation

### D1  Present climate

In this section, we present and discuss the ability of ARPEGE atmospheric model to represent the broad features of the
10  atmospheric general circulation around Antarctica. The winter (JJA) and summer (DJF) 500 hPa geopotentials and sea-level pressures (SLP) for ERA-I reanalyses and the ARP-AMIP simulation are presented in Fig. D1. In winter, it can be seen than ARPEGE reproduces quite correctly the 3 climatological minimum in SLP and the localization of the maximum of the South Polar vortex above the Ross Sea rather than on the South Pole. However, as already mentioned, the depth of the three SLP minimum and the meridional gradient around 50 to 60°S is underestimated. This remark is also valid in summer. It can also

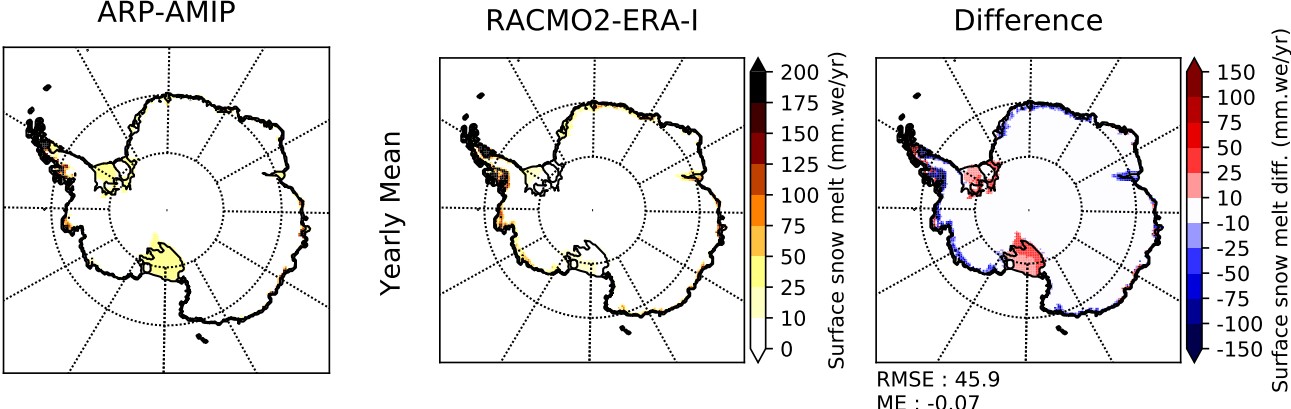

**Figure C4.** Yearly mean surface snowmelt (mm.we yr$^{-1}$ ) in ARP-AMIP (left), RACMO2-ERA-I (centre) and differences between the two models (right).

be noted that ARPEGE reproduces relatively correctly the displacement of the third SLP minima (Amundsen Sea Low) from eastern Ross Sea in winter to the Bellingshausen Sea in summer.

## D2 Consistency of the atmospheric model response

In this section, we briefly discuss the consistency of the response of the atmospheric model ARPEGE when forced by similar
SSC between present and future climate mentioned in the discussion. For the similarity of the SSC bias, see Fig. A1 and Fig. A2. This consistency of the atmospheric model response is considered as being the key for having similar climate signals between climate projections realized with or without bias-corrected SSC. In Fig. D2, the difference in SLP between ARP-NOR-20 and ARP-AMIP for the four climatological seasons and the corresponding difference for future climate (ARP-NOR-21-ARP-NOR-21-OC) are shown. It can be seen that there are few changes in the differences pattern between present and future climate
which is to be related with the minor differences in climate changes signal found for many variables in the experiment with bias-corrected and original NorESM1-M SSC. In Fig. D3, the same differences for the experiment performed with MIROC-ESM SSC are displayed. Here again, the pattern of the differences are very similar. We note however a tripole in the difference for future climate (ARP-MIR-21 - ARP-MIR-21-OC) in autumn (MAM), which was absent in the difference for present climate. This tripole can certainly be related to the tripole observed for the differences in precipitation and sea-level pressure change
signal observed in section 3.2.

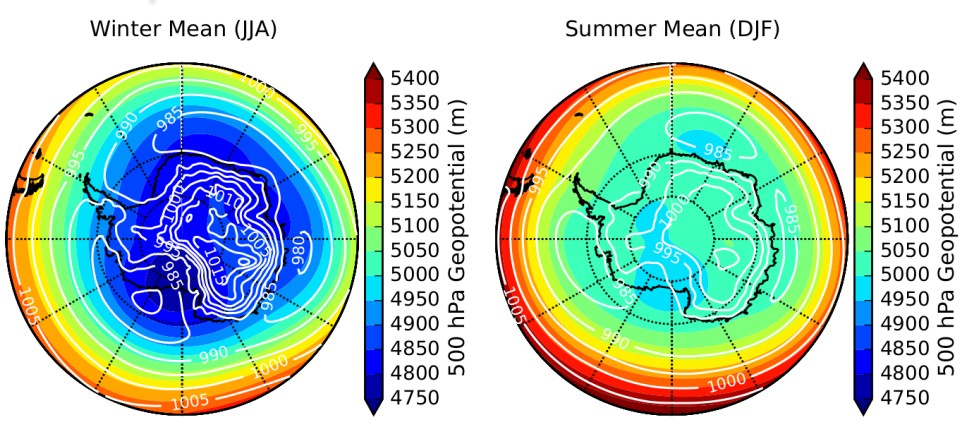

(a) ERA-Interim

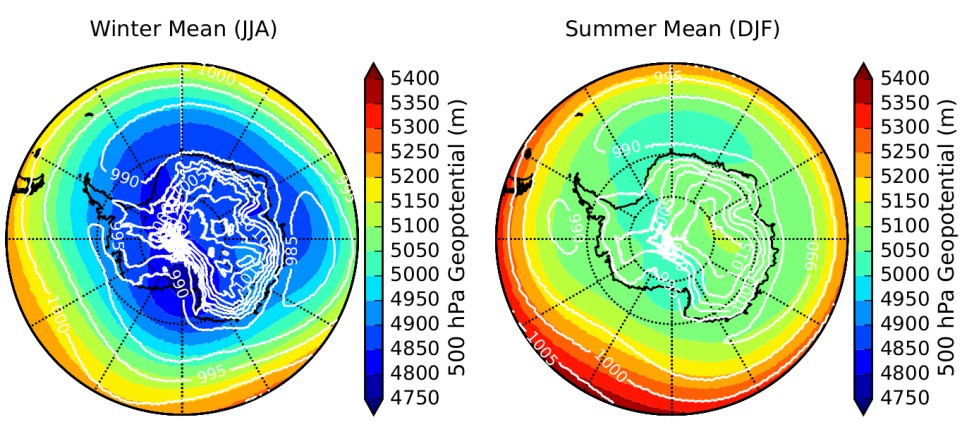

(b) ARP-AMIP

**Figure D1.** ERA-Interim (*top*) and ARP-AMIP(*right*) 500 hPa geopotentials (shadings) and sea-level pressures (white contour lines) in winter (*left*) and summer (*right*) for the reference period 1981-2010.

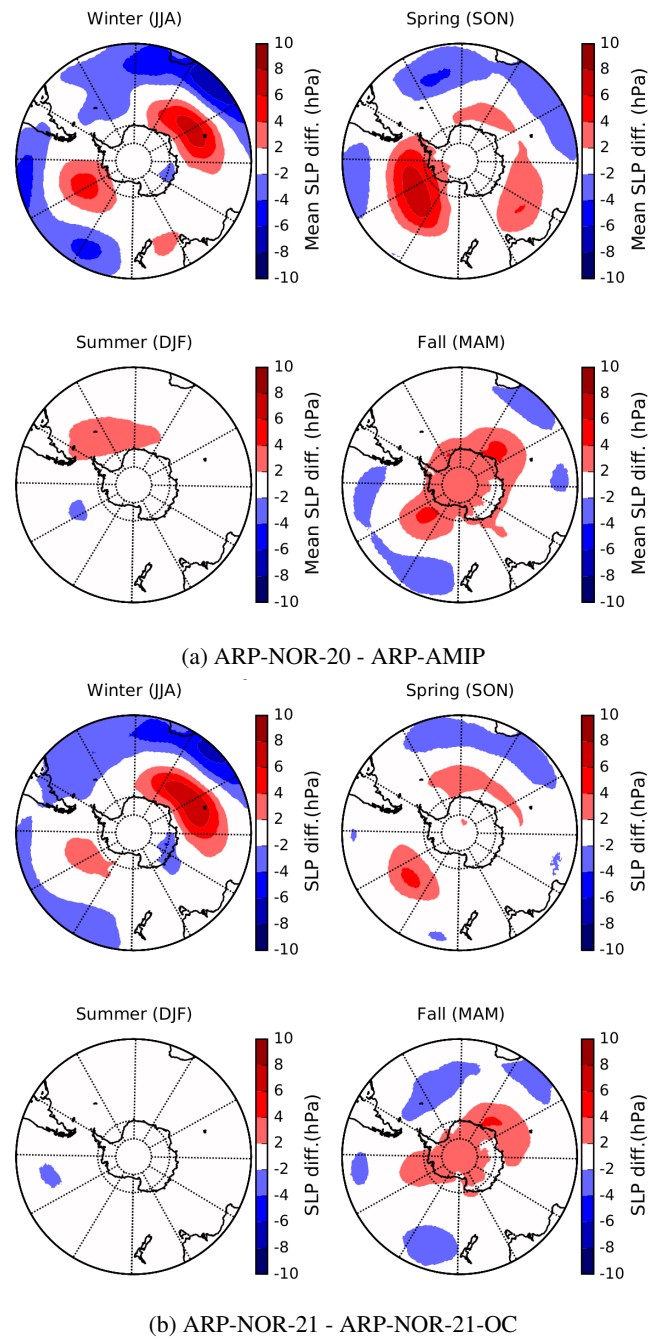

**Figure D2.** Difference between ARP-NOR-20 and ARP-AMIP for seasonal sea-level pressure (*top*) and corresponding differences for late 21[st] century, ARP-NOR-21 minus ARP-NOR-21-OC

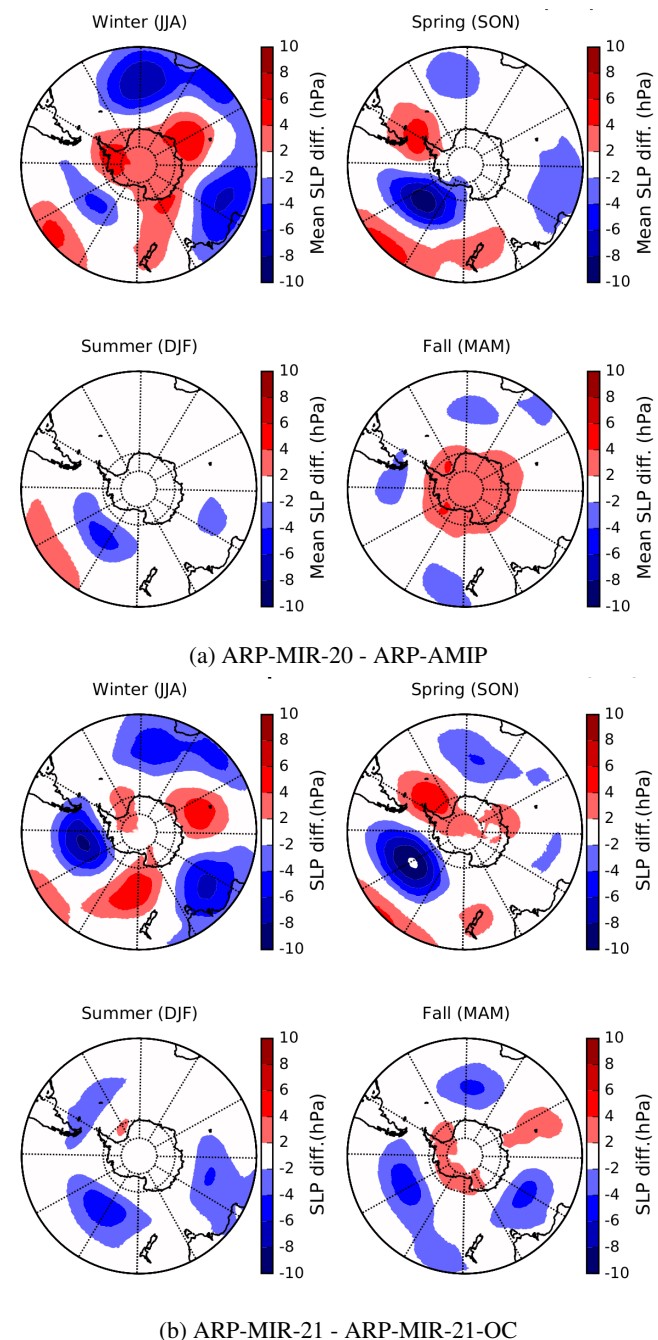

(a) ARP-MIR-20 - ARP-AMIP

(b) ARP-MIR-21 - ARP-MIR-21-OC

**Figure D3.** Difference between ARP-MIR-20 and ARP-AMIP for seasonal sea-level pressure (*top*) and corresponding differences for late 21st century, ARP-MIR-21 minus ARP-MIR-21-OC