# Peer review of "Effect of prescribed sea-surface conditions on the modern and future Antarctic surface climate simulated by the ARPEGE AGCM"

_The Cryosphere, 2018_

## Referee Comment (RC1) · Anonymous Referee #1 · 30 Jan 2019

This paper presents sensitivity studies with the atmosphere-only, stretched-grid GCM ARPEGE, forced by two present-day and strongly diverging end of 21st century sea ice and SST conditions from (bias-corrected) CMIP5 models. The results show that the Antarctic SMB is sensitive to Southern Ocean conditions, resulting from temperature and general-circulation changes. Although the paper contains some interesting results, it is very poorly written, contains factual errors, is and does not seem to come up with any clear answer to the problem posed in the title. I think it would require a very considerable effort from the authors to rewrite and strengthen the paper. I have decided not to focus on the language, but that doesn't mean that the paper needs a thorough check – it contains a lot of textual and grammatical errors! Instead, I will focus on (what I think are) the major issues with this paper, and hope the authors are able to improve the paper considerably. The only reason I decided not to reject is that I think the paper contains some interesting (but preliminary) results, but it will need to be thoroughly revised.

**Major issues (in order of appearance)**

Title: I think the title is a bit too general, and the paper does not really address it (see below for details). Something like: "Impact of two diverging scenarios of 21st century Southern Ocean surface changes on Antarctic surface climate and precipitation".

Abstract: the abstract needs a few introductory and concluding sentences, introducing the problem and motivation, and giving some concluding remarks ('what did this study find, in relation to the title?')

Surface mass balance: is only one term of the mass balance; importantly, not the SMB causes a decrease in sea level, but the change (increase) in SMB, assuming solid ice discharge doesn't change. Since SMB and discharge are intimately linked, it is incorrect to describe SMB as a negative term contributor to sea level rise.

P2, L28: …allowing the use of cloud-resolving atmospheric model configuration. I think you mean 'preventing' instead of 'allowing'?

P2, L33: *higher horizontal resolution leads to higher estimates of snow accumulation*. This is factually incorrect – actually, Genthon et al. (2009) suggest the opposite (see their Fig. 1). In addition, Lenaerts et al., 2017 do not find any significant impact of resolution on (integrated) SMB in the Amundsen region.

P3, L7: *RCM*. These random acronyms lead me to believe that the authors have been sloppy and have not sufficiently rechecked their manuscript prior to submission. Make sure these are defined when used for the first time.

P3, L18-29: This type of information does not fit in the introduction, it is far too detailed and should be moved to the methods.

Table 2: What are the units? What is the significance of these results, based on how much it varies in ERA-Interim over 1981-2010?

P7, L20: *9.5 Kelvin/km*. Where does this lapse rate come from? It would require a reference to back up this number.

P9, L8 and around: This temperature bias is highly concerning, and instead of simply removing these areas, I would advise the authors to try to explain (and remedy) this bias. My intuition is that ARPEGE is not well able to represent strong surface-based temperature inversions (which not be surprising as many climate models struggle with this). Also, these simulations will likely need to be redone with ice shelves (mind the spelling) considered in the land model – that will allow the authors to analyze the effect of changing ocean conditions on ice shelves (which are a super-important component of the Antarctic glacial system – and located closest to the ocean, so should be most sensitive!). In any case, the authors will need to come up with an explanation why the ice shelves are so warm in the model, will need to remedy that bias, and apply that to new simulations. The current bias is alarming, because there is no reason why this bias wouldn't apply to other regions on Antarctica – where this bias is potentially compensated for by other model biases (radiation, clouds, albedo,…)?

Table 3 is very poorly readable, enlarge and perhaps move to supplementary material. Again, don't forget to mention units. Same for Table 4.

P12, L12: this contradicts what was (falsely) mentioned in the introduction, as ARPEGE (the lower-resolution model) gives higher precipitation than MAR (the higher-resolution model)

P12, L18 and around: Runoff is the result of a complex interaction between atmosphere and snow conditions, and requires a sophisticated albedo and snow model, the latter which allows for percolation and refreezing of surface meltwater. The authors do not present any compelling evidence why the surface melt and runoff rates in ARPEGE are any realistic, which casts doubt on the reliability of simulated future melt and runoff rates. For example, Table 5 suggests that, on the grounded AIS, about one-third of the liquid water production (rain + melt) runs off in ARPEGE, which suggests that its snow model is not capable to retain and refreeze sufficient meltwater (for comparison: both MAR and RACMO2 produce almost no runoff with comparable liquid water production). I would therefore advise the authors to focus solely on precipitation and temperature, possibly surface melting (provided that the authors can show evidence of realistic surface melt patterns in the present-day simulation, compared to MAR for example), but refrain from analyzing future runoff changes.

Table 6: Are these changes significant at all? What is the present-day variability? What is the relative change instead of / next to the absolute changes?

Conclusions: a concluding paragraph/section is missing on the actual conclusion of this work. What is the uncertainty of Southern Ocean conditions on Antarctic SMB? What is driving it? What is the impact of changing SIC vs. SST? What are the driving forces of the change in Antarctic SMB – the thermodynamic (i.e. increase in surface temperatures) or

the dynamic (large-scale atmospheric circulation)? What is the impact of the radiative and turbulent fluxes? There are many open questions that the authors do not discuss, but that can be answered if the model simulations are analyzed in more detail.

**References**

Genthon, C., Krinner, G., Castebrunet, H., 2009. Antarctic precipitation and climate change predictions: horizontal resolution and margin vs plateau issues. Ann. Glaciol. 50, 55–60(6).

Lenaerts, J.T.M., Ligtenberg, S.R.M., Medley, B., Van de Berg, W.J., Konrad, H., Nicolas, J.P., Van Wessem, J.M., Trusel, L.D., Mulvaney, R., Tuckwell, R.J., Hogg, A.E., Thomas, E.R., 2017. Climate and surface mass balance of coastal West Antarctica resolved by regional climate modelling. Ann. Glaciol. 1–13.

---

## Editor Comment (EC1) · Christian Haas (Editor) · 6 Mar 2019

Dear authors,

unfortunately we have still not received a second review of your manuscript. In order to proceed swiftly and instead of recruiting and waiting for a second reviewer, I suggest to end the public discussion of your manuscript as soon as possible. Please provide your comments and replies to the one review received so far. We will then proceed with the review of your revision. I share many of the concerns of the first reviewer and therefore plan to send your revised manuscript out to two more reviewers eventually. Hopefully you agree with this procedure, and we look forward to receiving your comments. Thank

you and best regards

Christian Haas

---

## Author Comment (AC1) · 7 Mar 2019

Dear Editor,

Many thanks for taking care of the review process of the manuscript and for proposing a solution that would allow it to carry on despite the difficulty to find a second and third reviewer so far. We will respond to the comments received from Reviewer 1 and integrate his/her suggestions to improve the quality of manuscript as soon as possible and certainly no latter than 4th April as suggested in the e-mail I received.

Best regards,

Julien Beaumet

---

## Author Comment (AC2) · 15 Mar 2019

This paper presents sensitivity studies with the atmosphere-only, stretched-grid GCM ARPEGE, forced by two present-day and strongly diverging end of 21st century sea ice and SST conditions from (bias-corrected) CMIP5 models. The results show that the Antarctic SMB is sensitive to Southern Ocean conditions, resulting from temperature and general-circulation changes. Although the paper contains some interesting results, it is very poorly written, contains factual errors, is and does not seem to come up with any clear answer to the problem posed in the title. I think it would require a very considerable effort from the authors to rewrite and strengthen the paper. I have decided not to focus on the language, but that doesn't mean that the paper needs a thorough check – it contains a lot of textual and grammatical errors! Instead, I will focus on (what I think are) the major issues with this paper, and hope the authors are able to improve the paper considerably. The only reason I decided not to reject is that I think the paper contains some interesting (but preliminary) results, but it will need to be thoroughly revised.

**Authors' reply:** We thank the reviewer for accepting to review the manuscript and for doing so rapidly. For the language, we will thoroughly check the whole paper again and have it read by (at least) one native speaker.

Major issues (in order of appearance)

Title: I think the title is a bit too general, and the paper does not really address it (see below for details). Something like: "Impact of two diverging scenarios of 21st century Southern Ocean surface changes on Antarctic surface climate and precipitation".

**Authors' reply:** The reviewer is right. We will change the title in « Impact of two diverging scenarios of 21st century Southern Ocean surface changes on Antarctic surface climate » or something close to this formulation. This corresponds to the actual content of the paper which was not the case for the submitted version.

Abstract: the abstract needs a few introductory and concluding sentences, introducing the problem and motivation, and giving some concluding remarks ('what did this study find, in relation to the title?')

**Authors' reply:** We will modify the abstract, add some introductory and concluding sentences and adapt it to the main findings associated with this study and its title.

Surface mass balance: is only one term of the mass balance; importantly, not the SMB causes a decrease in sea level, but the change (increase) in SMB, assuming solid ice discharge doesn't change. Since SMB and discharge are intimately linked, it is incorrect to describe SMB as a negative term contributor to sea level rise.

**Authors' reply:** Ok, in order to re-conciliate these considerations, we propose to rewrite this sentence in the following way "Assuming no associated response of the glaciers dynamics, the increase of the ice-sheet surface mass balance is the only significant projected negative contribution to SLR... »

P2, L28: ...allowing the use of cloud-resolving atmospheric model configuration. I think you mean 'preventing' instead of 'allowing'?

**Authors' reply:** No, we in-deed meant 'preventing', but we propose to rephrase this sentence in the following way in order to hopefully make it less confusing : "*The marginal importance of atmospheric deep convection for Antarctic precipitation does not require to perform dynamical downscaling at very high resolutions and the use of a cloud resolving atmospheric model configuration is therefore not particularly relevant for Antarctic climate projection. However, the added value of higher horizontal resolutions, such as for instance CORDEX-like simulations (Giorgi et al., 2016) at 0.44°, with respect to driving climate projection at coarser resolution (1 to 2°) from the CMIP5 ensemble is significant in coastal regions".*

P2, L33: higher horizontal resolution leads to higher estimates of snow accumulation. This is factually incorrect – actually, Genthon et al. (2009) suggest the opposite (see their Fig. 1). In addition, Lenaerts et al., 2017 do not find any significant impact of resolution on (integrated) SMB in the Amundsen region.

Authors' reply: "The reviewer is right, this is indeed factually incorrect for present-day snow accumulation estimation. In Genthon et al. (2009), it is also found that resolution has no significant impact for model run at sufficiently high resolution (< 3°). Using 27kms and 5.5 kms set up of RACMO, Lenearts et al., (2018) for the Amundsen region and Lenearts et al., (2012) for Adélie Land indeed found that the area integrated surface mass balance and the coastal-inland precipitation gradient were not significantly changed. One of their conclusion is that 27 kms seems to be a sufficiently high horizontal resolution to represent the coastal-inland SMB gradient in West and East Antarctica. These conclusions are possibly no longer valid when we jump from 200 kms resolution used in CMIP experiments to 30-40 kms horizontal resolution used for instance in Cordex-like experiment, our study or the work from Lenaerts and others. However, the part of this sentence about climate projection is not incorrect as Genthon et al., 2009 found a strong sensitivity of projected Antarctic precipitations increase to resolution (higher increase for higher horizontal resolution) especially for resolutions below 2° (see their figure 2). Result from Agosta et al., (2013) who used LMDZ4 model at a horizontal resolution of 60 kms and downscaled these climate projections with SMiHil model at 15 kms agree with these findings. To our knowledge, there is no publication suggesting no or opposite effect of higher horizontal resolution on Antarctic precipitation increase in a warmer climate. To be factually correct about the effect of horizontal resolution on present-day and future changes in snowfall, and re-conciliate the findings of each study cited here above, we propose to rewrite this part of the article in the following way :

"For present-day climate, Lenaerts et al., (2016,2018) found no significant differences in areaintegrated SMB and coastal-inland snowfall gradient using 5.5 and 27 kms set up of RACMO model. Genthon et al., (2009) similarly found reduced impact of horizontal resolution when excluding very coarse (> 4°) model of the CMIP3 ensemble. For future climate projections however, much larger precipitation increases were reported when using climate model at higher horizontal resolutions (Genthon et al., (2009), Agosta et al., (2013)."

P3, L7: RCM. These random acronyms lead me to believe that the authors have been sloppy and have not sufficiently rechecked their manuscript prior to submission. Make sure these are defined when used for the first time.

**Authors' reply:** Ok, we have defined the RCM acronym at this place in the manuscript and checked carefully the introduction of new acronyms elsewhere.

P3, L18-29: This type of information does not fit in the introduction, it is far too detailed and should be moved to the methods.

**Authors' reply:**Ok, we will move the content of L18-29 and integrate it to the content of the "Data and Methods" section

Table 2: What are the units? What is the significance of these results, based on how much it varies in ERA-Interim over 1981-2010?

**Authors' reply:** Units are hectoPascals. We will perform some proper significance tests using the variability of sea-level pressure in Era-Interim, but these errors are likely to be significant as we plotted the significance (not shown) of ARPEGE sea-level pressure bias with respect to ERA-Interim and it is significant almost everywhere (at p=0.05) South of 20°S.

P7, L20: 9.5 Kelvin/km. Where does this lapse rate come from? It would require a reference to back up this number.

**Authors' reply:** A dry adiabatic lapse rate of 9.8 K.km-1 (there was indeed a small typo here) is used for instance in Bracegirdle and Marshall (2012) to correct surface temperature from meteorological reanalysis in order to compare them with in-situ observations in Antarctica. We will refer to this publication to justify to use of this lapse rate.

P9, L8 and around: This temperature bias is highly concerning, and instead of simply removing these areas, I would advise the authors to try to explain (and remedy) this bias. My intuition is that ARPEGE is not well able to represent strong surface-based temperature inversions (which not be surprising as many climate models struggle with this). Also, these simulations will likely need to be redone with ice shelves (mind the spelling) considered in the land model – that will allow the authors to analyze the effect of changing ocean conditions on ice shelves (which are a super-important component of the Antarctic glacial system – and located closest to the ocean, so should be most sensitive!). In any case, the authors will need to come up with an explanation why the ice shelves are so warm in the model, will need to remedy that bias, and apply that to new simulations. The current bias is alarming, because there is no reason why this bias wouldn't apply to other regions on Antarctica – where this bias is potentially compensated for by other model biases (radiation, clouds, albedo,...)?

**Authors' reply:** The reviewer is right in stating the fact that this temperature bias is concerning. We found it concerning as well and we tried unambiguously to identify its origin. First, we verified if ice shelves are indeed treated as land surface in the model, we plotted surface sensible and latent heat fluxes (see figures below) as well as surface albedo (SWU/SWD) and from this point of view nothing is abnormal and it compares reasonably with the same fluxes in MAR. The reviewer is also right in it is intuition that the warm bias over ice shelves (in winter) comes from ARPEGE lack of skills to represent very stable boundary layer and associated strong near-surface temperature inversions (as many climate models do). To investigate this, we plotted the difference between air temperature at 20 meters and surface temperature. We can see that the magnitude of the near-surface inversion compares reasonably well over most of Antarctica except over the ice shelves where the pattern (too weak inversion in winter and too large in summer) seems to be seasonally and spatially consistent with the biases (or difference) in near-surface temperature with MAR. The seasonality of the biases is the same over the high Antarctic Plateau. The reviewer is also right in saying that in other parts of Antarctica, ARPEGE lower skills for boundary layer are slightly compensated for by

other biases (or difference with respect to MAR). In another ARPEGE experiment, in which we corrected atmospheric general circulation using nudging towards reanalyses (other paper in prep.), the warm bias over the High Antarctic Plateau increased slightly (1-2 K) as result of a decrease of a negative downward LW bias in the ARPEGE-AMIP experiment, but this warm bias in winter with respect to MAR and observations (3 - 5K) is in any case not much higher than what many other GCMs or even meteorological reanalyses are showing for the near-surface temperature of the Antarctic Plateau. It seems that the exceptional characteristics of the large ice shelves (extremely large and flat surfaces with few roughness) highlight more than anywhere ARPEGE lack of skills for extremely stable boundary layers.

Moreover, we draw the attention on the fact that a part of the large difference (10-12K) between ARPEGE and MAR over ice shelves in winter also comes from a cold bias of the MAR model as in evaluation against 12 stations over the Ross Ice Shelf her (https://zenodo.org/record/1256079#.XIuPd5zjIUF), C. Agosta found a -2.8K cold bias. This can also be seen in our comparison over the smaller ice shelves of the Dronning Maud Land region, as the comparison with MAR-ERA-I (Figure 4, left) suggest a large (5 to 9 K) warm "bias" in ARPEGE, while the warm bias of ARPEGE with respect to Halley and Neumayer weather stations are respectively only +1.2K and +0.9K. We also draw the attention on the fact that warm "biases" in winter over ice shelves is slightly reduced over Ross Ice Shelf and almost completely disappears over Ronne-Filchner Ice Shelf when RACMO2 is taken as reference (see Fig. below). This highlight how much large discrepancies can still exist over ice shelves, even between polar-oriented climate models regularly used as reference for Antarctic surface climate and mass balance such as MAR and RACMO2.

We precise that it is virtually impossible for us to redo all our ARPEGE experiment as our collaborator and co-author of the paper Michel Déqué is now retired and we therefore currently have no available computer time on the Météo-France supercomputer. Besides, fixing ARPEGE issues for stable boundary layer is a work beyond the scope of this paper and is actually the subject of a PhD thesis currently undertaken at Météo-France. Even if we agree that this bias is concerning, we think that many climate models or even meteorological reanalyses (see Freville et al., 2014, Bracegirdle and Marshall, 2012) show biases of Antarctic surface temperatures (High Plateau or ice shelves) that are the same order of magnitude as the warm bias over large ice shelves in our ARPEGE simulations. This, however did not prevent these data to be published and unfortunately sometimes widely used. So, we agree on the following for the future versions of the manuscript: avoiding to hide these biases, trying to be more explicit about their origin and warn potential users over ARPEGE reduced skills over ice shelves, being more critical about the skills of models (MAR and RACMO2) used to evaluate ARPEGE. Unfortunately, restarting the simulation while remedying the bias over the ice shelves will not be possible. To evaluate with reduced uncertainties the impact of climate change over Antarctic ice shelves, we propose to use our ARPEGE future projections to drive regional climate models (e.g. MAR or RACMO2) that are more skilled for ice shelves surface climate, which is also currently a work in progress.

---

## Referee Report (RR1)

Review Beaumet

This paper presents a sensitivity study of ARPEGE-AGCM, a global climate model at a stretched grid such that it has a horizontal resolution over the Antarctic ice sheet comparable to typical regional climate models. This implies a high and original potential for high-resolution projection studies for this region, and its connections to global climate. However, the model incorporates far too many model deficiencies (most importantly the ice shelf biases (if you look at the impacts of SSC, then ice shelves are so important!) and the limited (surface) snow model) that in my honest opinion have to be solved first in order to warrant publication. In addition, the paper is already in its second review round and still does not fully answer the question raised in the title and is still poorly written.

I will not reject the paper as I understand that new simulations will be extremely difficult to perform. I do think that in order to warrant publication, more substantial results and conclusions should be presented and the outline of the paper should be restructured and have a stronger structure. If the main aim of this paper is to be an evaluation paper (as I think in principle this is what this paper is about) this should be the main focus, with a more substantial analysis evaluation and a comparison with other GCMs at coarser resolutions (as to strengthen the main advantage of ARPEGE, its resolution). If the approach is still to be to investigate the effect of changing SSC, I want to see more of it and a more thorough discussion of its implications.

Even though I did not have time or the motivation to tackle the text and do minor corrections to the sloppy writing, I summarize some of my main issues with the paper below:

P1, abstract, l7: The abstract should be readable and understandable without reading any of the other text. For me it was not directly clear what is meant with "diverging SSC". Do you mean two extreme cases? Reword this sentence.

P1, abstract, l15-16: Same as above, what is meant with quantile mapping; completely unclear to me from the abstract alone. Make it more simple.

P1, abstract, l19-21: Same as above, unclear from just the abstract. What is bias-corrected SSC?

P2, abstract, l1-2: Is this not well known already? Of course circulation is extremely important (for what, by the way?)

P3, l23: "some advantages". Which advantages? I want to know precisely why the model setup is preferential over others.

P3, l32: Again, I still fail to properly understand "bias-corrected SSC".

P4, l14: -> climate of the Antarctic continent.

P5, Figure 1: Maybe highlight the models that you are going to use

P7, paragraph 3.1: This seems like a section that should be in data-methods, not in results.

P9, Figure 2: In the text you refer to latitudes, but no latitudes are shown on this figure. I had to check twice to see what you meant with 40 degrees S.

P10, l7: name the stations on the map that you refer to. Maybe a scatter plot also is more clear than this plot as the range of the colorbar is quite small.

P10, l10: "site effect". What site effect? Do you think that a GCM can be compared with in-situ locations in general?

P12, l7: What is "2.8 interannual standard deviation?

P13, figure 5: I fail to properly see the significance lines; strange colouring is used for this figure.

P21,l21: I don't understand this line. Thermodynamic is not related to circulation patterns?

Discussion: The discussion looks to me like a long monologue without any goal or clear structure and it is way too long. I therefore propose to restructure the results and discussion section and try to work towards the main conclusion of the study (which to me is still too vague). Start with a "Results: Evaluation" section, and end with a "Results: effects of SSC", or something like that. Now, I had to reread several times before I understood your structure.

---

## Author Response (AR2)

**Reviewer 1 – Jan Lenaerts**

I thank the authors for taking the time to address my comments and for revising this paper considerably. I think it has improved tremendously, and I applaud the authors for their effort. However, I have a few remaining comments that should be addressed, which I briefly highlight below.

We thank the reviewer for reviewing the paper for a second time and for measuring the effort we made to improve the paper during the previous stage of the rewiew process.

P1, L9: Antarctic is capitalized (this error returns a few time)

Authors' reply: "Following this website (https://arcticisms.com/2011/04/27/to-capitalize-or-not-to-capitalize-that-is-thequestion/), Antarctic should always be capitalized. But perhaps, we (and this website) are wrong. Surely, TC has its own well-established rules about this case that we will have to follow later."

P2, L9: uncertainties in

Authors' reply: "Ok, comment taken into account"

P3, L21: use one number consistently, either 35 or 40 km

Authors' reply: "Ok, comment taken into account"

P6, L12: moisture fluxes

Authors' reply: "Ok, comment taken into account"

P6, L13: sea ice

Authors' reply: "Ok, comment taken into account : all occurrences of 'sea-ice' in the paper have been replaced by 'sea ice' but we kept 'sea-ice' when it is an adjective (e.g. sea-ice concentration)."

P6, L20: I would use 'soil/snow' or something equivalent

Authors' reply: "Ok, comment taken into account, we use now 'soil and snowpack'."

P7, L3: below = southward of / poleward of

Authors' reply: "Ok, comment taken into account."

P7, L16: although = however

Authors' reply: "Ok, comment taken into account."

Section 3.1.1: more quantitative description is required here. Associate qualitative assessments such as 'underestimation', 'similar to', etc. with actual numbers.

Authors' reply: "Ok, we have added more quantitative description and trimmed a little bit this section as well."

P8, L16: temperatures are higher, not warmer

Authors' reply: "Ok, comment taken into account."

Table 4: this is my largest issue with the new paper – the caption of the table mentions an area of 13.4 million km2 for the grounded AIS, and 12.3 million km2 a few lines below that. When comparing AIS integrated numbers, it is extremely important to use a common mask, such that the integrated AIS area is equivalent between all of the estimates.

Authors' reply: "Ok, we recomputed SMB and other variables statistics using the MAR RCM's grounded ice mask for each model (grounded AIS = 12,4 million km2). For consistency, we of course also did so for climate projections and changed the text accordingly. The main change in these recomputed statistics is that now, total precipitation over the GIS in ARPEGE-AMIP is very close to estimates from MAR and RACMO2, while estimates of the SMB are lower due

to excessive sublimation and run-off in ARPEGE. For future climate projections, the conclusions about changes in integrated SMB and other variables over the GIS are mostly unchanged."

P14, L3: a shift of the westerly jet would not necessarily be associated to a positive phase of the SAM – the SAM simply illustrates the surface pressure differences between 40 and 65 degrees South, and hence is more a manifestation of the meridional pressure /temperature gradient and associated strength of the westerlies. As an example, the SAM does not 'know' if the westerly jet maximum is at 55 degrees or 52 degrees South.

Authors' reply: "Indeed, the reviewer is correct: a positive phase of the SAM is much more tied to a higher pressure gradient between mid and high latitude than to a poleward shift of the surface westerly jet maximum. However, both (increase in the pressure gradient and poleward shift) are the expected consequences of enhanced greenhouse gases forcing for the end of current century. In order to make this sentence less confusing and more factually correct, we rephrase it in the following way: 'This corresponds to a strengthening of the mid to high latitude pressure gradient (positive phase of the SAM) and a poleward shift of the circum-antarctic low pressure belt towards the continent, which are generally the expected consequences of 21st century climate forcing (Kushner et al., (2001), Arblaster et al., (2006))...'"

Table 6 caption: century is missing

Authors' reply: "Ok, comment taken into account"

**Table 6: How can the numbers be significantly different if the uncertainties overlap?**

Authors' reply: "To determine if a one sample b is different from sample a (independent mean), we use the following test: the difference between the mean of a and b must be greater than  $(1.96*(\sqrt{0.5(STDa+STDb}))*\sqrt{2}) / \sqrt{(n-2)})$  where STDa and STDb are the standard deviation of sample a and b and n the number of individuals in our samples (here 30 because 30 years). In table 6, for the only significant difference, the test above yields 0.733, which is smaller than the difference 0.8. So we concluded that it is significant at p=0.05 even though it is close to the significance threshold."

P17, L4: For the experiment Authors' reply: "Ok, comment taken into account"

P21, L25: something is wrong with the reference Authors' reply: "Ok, comment taken into account"

P22, L29: circumspection = caution Authors' reply: "Ok, comment taken into account"

Section 4.3: this is a very lengthy discussion – consider trimming.

Authors' reply: "We have trimmed the section 4.3 and the discussion in general which was also a suggestion of reviewer 2."

**Reviewer** 2**

This paper presents a sensitivity study of ARPEGE-AGCM, a global climate model at a stretched grid such that it has a horizontal resolution over the Antarctic ice sheet comparable to typical regional climate models. This implies a high and original potential for high-resolution projection studies for this region, and its connections to global climate. However, the model incorporates far too many model deficiencies (most importantly the ice shelf biases (if you look at the impacts of SSC, then ice shelves are so important!) and the limited (surface) snow model) that in my honest opinion have to be solved first in order to warrant publication. In addition, the paper is already in its second review round and still does not fully answer the question raised in the title and is still poorly written. I will not reject the paper as I understand that new simulations will be extremely difficult to perform. I do think that in order to warrant publication, more substantial results and conclusions should be presented and the outline of the paper should be restructured and have a stronger structure. If the main aim of this paper is to be an evaluation paper (as I think in principle this is what this paper is about) this should be the main focus, with a more substantial analysis evaluation and a comparison with other GCMs at coarser resolutions (as to strengthen the main advantage of ARPEGE, its resolution). If the approach is still to be to investigate the effect of changing SSC, I want to see more of it and a more thorough discussion of its implications.

Even though I did not have time or the motivation to tackle the text and do minor corrections to the sloppy writing, I summarize some of my main issues with the paper below:

Authors' reply: We thanks the reviewer for accepting to review the paper. As it is relatively new to use ARPEGEclimate model in general and the high\_-resolution configuration used this study in particular to study the Antarctic climate, this naturally calls for a general evaluation of the model skills for the high southern latitude. This was however not the first aim of this study. The evaluation of the model and the study of the effect of changing SSC could have been tackled into two different manuscripts, however, at this stage of the review process, it is unrealistic to move toward this direction.

We specify that Antarctica and the high southern latitudes in general is probably the area of the globe where the least number of continuous and reliable meteorological observations records exists. As a consequence, for some climate variables, the evaluation of a freely evolving climate model is quite challenging. Despite this fact, and although some supplementary evaluation of the ARPEGE model could be added to the paper (which would make it very long), the manuscript already contains a substantial amount of evaluation of the model: we evaluated atmospheric general circulation (sea-level pressure and latitudinal wind profiles) using ERA-Interim reanalyses, the most reliable reanalyses for the high southern latitudes, the near-surface temperatures using in-situ data and state-of-the-art polar-oriented RCM MAR and surface mass balance and its component using MAR and RACMO2. In the supplementary material, there is also a comparison between ARPEGE and MAR for most components of the surface energy balance (latent and sensible heat flux, -downward longwave radiation), for the strength of the near-surface temperature inversion and for the surface albedo. We also performed an evaluation of 10m wind speed using READER weather station and MAR RCM, but we think that adding this to the manuscript is not relevant. Nevertheless, we performed a supplementary evaluation of ARPEGE precipitation by using a model independent data set : the CloudSAT snowfall climatology from Palerme et al., (2014). However, the results of these comparisons have to considered with a lot a caution because the CloudSAT climatology is only available for 2007-2010 and is representative for snowfall about 1200 m above the surface. This why (togheter with the purpose of limitating the length of the main part of the manusctipt) these results are only shown in the supplementary material (see section C2). This section is called in the evaluation subsection of the data-method section main part of the manuscript (page 7, line 20): "We also performed an evaluation of ARPEGE snowfall rates using a model independent data set such as the CloudSAT climatology for antarctic snowfall \citep{Palerme2014}. However, because this data set is only available for a very short period of time (2007-2010) and is representative of snowfall rates about 1200 m above the surface, the results from this comparison have to be considered with extreme caution and are therefore only shown in the supplementary material (see \ref{secCloudSAT})."

The temperature bias of ARPEGE over large ice shelves when compared to MAR RCM seems indeed at first order quite concerning. However, we think that we demonstrated quite convincingly in the supplementary material that the importance of this bias should be put into perspective. Firstly, about a third of the MAR-ARPEGE discrepancy is actually due to a MAR cold bias of large ice shelves. Secondly, we have shown that we understand the origin of the bias (misrepresentation of boundary layer processes) and that ARPEGE (when compared to MAR) shows consistent values for the fluxes of the surface energy balance and for the surface albedo. Finally, we also showed that large discrepancies over ice shelves exist between the polar-oriented RCMs MAR and RACMO2, both considered as references for the Antarctic climate, and that the wintertime near-surface temperature errors of ARPEGE over large ice-shelves are greatly reduced when compared to RACMO2. In fact, Antarctic near-surface temperature in many climate models or even climate reanalyses shows substantial errors (Bracegirdle et al., 2012; Freville et al., 2014) that are sometimes even

larger than errors shown by ARPEGE over large ice shelves. This did not prevent this simulation or data set to be published or even sometimes widely used, unfortunately.

Where relevant, we have added some references or elements of discussion of the added value of the high spatial resolution of our simulation, following your recommendations and we have restructured the discussion section of the paper in order to make the structure of the paper easier to handle."

P1, abstract, l7: The abstract should be readable and understandable without reading any of the other text. For me it was not directly clear what is meant with "diverging SSC". Do you mean two extreme cases? Reword this sentence.

Authors' reply: We rephrased this part of the abstract in the following way : « This study evaluates the Antarctic surface climate simulated by a high-resolution atmospheric model, and assesses the effects on the simulated Antarctic surface climate of two different sea surface condition (SSC, i.e. sea surface temperature and sea ice concentration) data sets obtained from projections with coupled climate models. The two coupled models from which SSC are taken, MIROC-ESM and NorESM1-M, simulate future Antarctic sea ice trends that are at the opposite ends of the CMIP5 RCP8.5 projections range. »

P1, abstract, l15-16: Same as above, what is meant with quantile mapping; completely unclear to me from the abstract alone. Make it more simple.

Authors' reply: Ok, we rephrased this sentence in the following way : «For the 2071-2100 period, SSC from the same coupled climate models forced by the RCP8.5 emission scenario are used both directly and bias-corrected with an anomaly method."

P1, abstract, l19-21: Same as above, unclear from just the abstract. What is bias-corrected SSC?

Authors' reply: The acronym SSC and what it refers to (sea surface temperature and sea ice concentration) is introduced in the abstract. In the correction above, we specify that the bias-correction method is based on a anomaly method. Explaining more precisely how the bias-correction method works, or what an anomaly method is, is beyond the scope of this abstract and would make it too long. Interested readers who want to know more precisely about how the bias correction method used, as well as those who do not know what an anomaly method is, are invited further away in the paper to refer to the Beaumet et al., (2019) paper published in GMD, where different bias correction methods (including the one used here) are presented and compared.

P2, abstract, 11-2: Is this not well known already? Of course circulation is extremely important (for what, by the way?)

Authors' reply: The point we wanted to make, as mentioned by Bracegirdle and al. (2015), is that errors of Southern Hemisphere general circulation in atmospheric models have as much influence on the projected Antarctic climate change as the changes in sea surface conditions (SSC), although SSC were shown to have much more influence on the evolution of the Antarctic climate than the additional forcing coming from greenhouse gases (Krinner et al., 2014). The reviewer is correct is mentioning that this is not new, so we rephrase this phrase in the following way: "This confirms that the errors in the representation of the South Hemisphere general circulation in the atmospheric models are also determinant for the diversity of their projected late 21st century antarctic climate change."

P3, 123: "some advantages". Which advantages? I want to know precisely why the model setup is preferential over others.

Authors' reply:One of the main advantages is that AOGCMs, as opposed to RCMs, do not depend on the reliability of the driving GCM for the representation of the atmospheric general circulation of the region of interest. Therefore, we specify this in the sentence mentioned: "This method has some advantages over the more commonly used limited-area RCM method which depends, for future climate projection, on the quality of the representation of the climate of the region of interest by the driving GCM used at lateral boundary" Although this is one advantage, together with other advantages mentioned further in the text, we do not consider (and do not state in the manuscript) that the use of stretched grid AGCMs for future climate projections is objectively better and should be privileged over the classical method using RCM driving by GCMs at their lateral boundaries to generate future climate projection. Both methods have their advantages and inconvenients and could be for instance combined.

P3, l32: Again, I still fail to properly understand "bias-corrected SSC".

Authors' reply: Hopefully, it has become clearer with the modification brought to the abstract and to the introduction. More precise explanation are given in the method section, and interest readers are invited to refer to the Beaumet et al., (2019) paper in GMD, which describes precisely the method used as well as other methods for the bias correction of SSC. We think that unfortunately a more in-depth and detailed description of how works bias correction of sea surface condition for climate projection with atmospheric model is beyond the scope of the introduction of the paper.

P4, l14: -> climate of the Antarctic continent.

Authors' reply: Ok, comment taken into account.

P5, Figure 1: Maybe highlight the models that you are going to use P7,

Authors' reply: Ok, we highlight in red in Figure 1 the models used in the study.

paragraph 3.1: This seems like a section that should be in data-methods, not in results. Authors' reply: Ok, we moved this section and created a new sub-section at the end of the data-methods section.

P9, Figure 2: In the text you refer to latitudes, but no latitudes are shown on this figure. I had to check twice to see what you meant with 40 degrees S.

Authors' reply: We have added the value of the parallels drawn on Figure 2 in order to make cross-reading with the text easier.

P10, 17: name the stations on the map that you refer to. Maybe a scatter plot also is more clear than this plot as the range of the colorbar is quite small.

Authors' reply: We have added the name of the stations used on the right panel of Figure 4. Readers are now not required any more to go to the map with station names previously put in the supplementary material in order to locate the stations used, so we deleted the map in the supplementary material. A scatter plot could have been relevant, but the paper already contains a considerable number of figures and requires the readers to know about the locations of the Antarctic weather stations. Readers interested by more precise values for the comparisons between ARPEGE and in-situ weather stations can refer to the Table in the supplementary material.

P10, 110: "site effect". What site effect? Do you think that a GCM can be compared with insitu locations in general?

Authors' reply: "By site effect, we meant the effect of local topography or shoreline configuration that influences the records of in-situ weather stations. Whether or not output from a GCM can be compared to data from local point measurements depends greatly on the resolution at which the climate model is used, and on the local variability of the topography and of the land properties. In Antarctica, the resolution of 35 kms is likely enough to represent the surface climate of the Plateau and upper parts of the ice sheet, where it is actually more the quality of the boundary layer scheme that challenges the representation of the surface climate of in-situ weather station. Over coastal areas, the variation of the topography and of surfaces properties cannot be captured by the GCM at the resolution used and comparison with in-situ measurements from these areas is indeed challenging. We modified "site effect" by "effects of the local topography" in the text".

P12, l7: What is "2.8 interannual standard deviation?

Authors' reply: We changed this to "...standard deviation of the annual mean" where relevant.

P13, figure 5: I fail to properly see the significance lines; strange colouring is used for this figure.

Authors' reply: We use colorblind compatible colormap for precipitation and surface mass balance diffences map as recommended. Over this colouring, cyan and pink for significance lines are among the easiest colour to distinguish. For sublimation, areas where the difference is "significant" are quite small and close to the shoreline, so there is not much

we can do. For the final version of the figure, we can possibly slightly improve them by increasing the smoothing and thickening a bit the significance lines."

P21,l21: I don't understand this line. Thermodynamic is not related to circulation patterns?

Authors' reply: "Here the thermodynamic component refers to the fact that for the same circulation pattern, the amount of heat or moisture that can be transported over some areas can change significantly in a warmer climate (due to for instance higher SST or reduced sea ice cover). This opposed to the "dynamic" component, which is the result of the change in relative frequencies of circulation patterns (e.g. Driouech *et al.*, (2008), Krinner *et al.*, (2014))."

Discussion: The discussion looks to me like a long monologue without any goal or clear structure and it is way too long. I therefore propose to restructure the results and discussion section and try to work towards the main conclusion of the study (which to me is still too vague). Start with a "Results: Evaluation" section, and end with a "Results: effects of SSC", or something like that. Now, I had to reread several times before I understood your structure.

Authors' reply: "We have reorganized the discussion into two main section as you suggested. We have trimmed the different sections of the discussion in order to make them more easy to read and in order to emphasize the most important results of the study. Where relevant, we have added some references to the presence or not of added value of the high spatial resolution, and discussed it in the light of previous studies such as you mentioned at the beginning of your report. The large number of changes applied to this section in response to this comment makes it impossible to list these changes individually here. "

[revised manuscript text omitted]
 etmospheric encoded in each simulation has also been eccentered end explored and explored for the formation.

The mean atmospheric general circulation in each simulation has also been compared and evaluated against ERA-I by analyzing the latitudinal profile of the 850 hPa eastwards zonal mean eastward wind component (referred to as westerly winds in the

30 following)<del>latitudinal profile</del>, as well as the strength (m/s) and position (°Southern latitude) of the zonal mean westerly wind maximum (Fig. 3). In this figure, results are only presented for the annual average, as the differences between simulations or

**Table 2.** Seasonal root mean square error (RMSE, in hPa) on mean SLP South of 20°S with respect to ERA-Interim for the different ARPEGE simulations over the 1981-2010 period. Each error is significant at p=0.05

| Simulations | DJF | MAM | JJA | SON |
|-------------|-----|-----|-----|-----|
| ARP-AMIP    | 3.3 | 2.7 | 3.1 | 3.0 |
| ARP-NOR-20  | 3.5 | 4.3 | 4.8 | 4.6 |
| ARP-MIR-20  | 3.2 | 4.0 | 4.6 | 3.2 |

with respect to ERA-I do not depend much on the season considered (*not shown*). ARP-AMIP and ARP-MIR-20 are closer to ERA-I when the westerly winds maximum strength is considered . The with an underestimation of this maximum about 1.5 m.s-1s. The equatorward bias on the position of the westerly wind maximum is closest to ERA-I 1.6° in ARP-NOR-20. With respect to , while it is up to 3 to 5° in ARP-AMIP , ARP-NOR-20 displays a much weaker and polewards shifted surface westerly wind maximum while and ARP MIP 20 is characterized by a lower latitude westerly wind maximum of comparable

5 westerly wind maximum, while and ARP-MIR-20is characterized by a lower latitude westerly wind maximum of comparable strength.

**3.1.2 Near-surface Temperatures**

Screen level (2 m) air temperatures ( $T_{2m}$ ) from ARP-AMIP simulation are compared to those from MAR-ERA-I simulation and READER data base in winter (JJA) and summer (DJF) for the reference period 1981-2010 (Fig. 4). In this analysis, stations

10 from the READER data base for which less than 80% of valid observations were recorded for the reference period were not used for the computation of the climatological mean. Altitude differences between corresponding ARPEGE grid point and stations have been accounted for by correcting modelled temperatures with a 9.8 K km-1 dry adiabatic lapse rate, such as done for instance in Bracegirdle and Marshall (2012). Errors on of the  $T_{2m}$  in ARP-AMIP simulation for each weather station and each season are presented in the supplementary material (Table B1)as well as a map showing the location of these stations (Fig.

The ARP-AMIP  $T_{2m}$  are much warmer than MAR-ERA-I on the ridge and the western part of the Antarctic Plateau in winter as well as on on the large Ronne and Ross ice shelves. Consistently with its atmospheric circulation errors in this area, ARPEGE is colder than MAR-ERA-I on the Southern and Western part of the Antarctic Peninsula, especially in winter. We can also mention a moderate (1 to 3 K) but widespread warm bias on the slope of the EAP and on the west side of the West

20 Antarctic Ice Sheet (WAIS) in summer. Except for some coastal stations of East Antarctica,  $T_{2m}$  errors in the ARP-AMIP simulation are very similar in the comparisons with MAR-ERA-I and READER data base.

Considering errors on near-surface temperatures of the Antarctic Plateau as large as 3 to 6 K for ERA-I reanalysis in all seasons (Fréville et al., 2014), skills of the ARP-AMIP simulation in this region is are comparable to those of many AGCM or even climate reanalyses. The systematic error for Amundsen Scott station is for instance not significant at the p=0.05 level

15 <del>??).</del>.

25 in all seasons but 5% level in any season except autumn (MAM). The large discrepancies between ARPEGE and MAR over large ice shelves are further investigated in section the appendix B1. Although a part (3-5K) of this large discrepancy in winter

(a) ARP-AMIP

---

## Author Response (AR3)

**Editor Decision: Publish subject to minor revisions (review by editor)** (05 Sep 2019) by Christian Haas**

Comments to the Author:

Dear authors,

thank you for revising your manuscript and for providing replies to the reviewer's comments. I agree that you have mostly taken them into account, but I would like to ask you to further improve the manuscript to make it more readable and to include the reviewer's considerations even more.

**Authors' reply: We thanks the editor for rapidly taking care of the review process of the manuscript.**

Therefore, please consider the following:

Please improve and shorten the abstract further, and make it more readable. A definition of SSC should follow after the first mentioning (i.e. in the first paragraph; avoid using parentheses). Also I agree that you should shortly define what bias correction is (or what the bias is) and what the anomaly method is. That can be done with a few words or a half-sentence at most.

Authors' reply: We modified the abstract in order to make it shorter and more readable and we added a short definition for SSC in the 1st paragraph of the abstract as you suggested and for the bias correction method further in the abstract « ...both directly and bias-corrected with an anomaly method which consists in adding the future climate anomaly from coupled models projections to the observed SSC with taking into account the quantile distribution of these anomalies. »

We also gives further information about the bias-correction method for sea ice in the Sea Surface Conditions subsection from Data and Methods (2.1) : « The bias-correction methods used for SST ans SIC mostly relies on anomalies methods, which consists in adding the anomalies coming from a coupled model projection to the observed SSC while taking into account the quantile distribution of these anomalies. Besides, the analog method for sea-ice recombines analog candidates from a library of observed and simulated SIC maps in order to reproduce SIE and sea-ice area computed using the anomaly method. »

Please take the reviewer's comments into account more explicitly, by including some of your replies explicitly in the text. This applies e.g. for the discussion of significance in Table 6 (reviewer 1), or the discussion of Thermodynamics (reviewer 2 comment about page 21). There are other replies that can be better included in the text as well to share your opinions not only with the reviewers but with all readers should they have similar questions.

Authors' reply: We add the information about the significance test at the end of the section 2.3 « Model Evaluation » in « Data and Methods » (P7 L25-30).

We modified slightly the discussion about Thermodynamics in order to make it more understandable and added some references.

We add the following sentence to the discussion concerning comparison of GCM output with in-situ observations in order to better take into account the reply to reviewer 2 (P11 – L17-19) : "Contrary to the continent's interior, the average 35 kms horizontal resolution used in this study is insufficient to capture many local topographic features of the coastal areas and of the AP, which challenges the comparisons with \textit{in-situ} measurements in these areas."

Following our reply to reviewer 2 to concerning the advantages of the use of AGCMs, we added the following sentence to the section 4.2.3 of the discussion (P25 L4): "The use of stretched grids AGCMs and polar-oriented RCMs to downscale future climate projections for Antarctica comports

their own assets and drawbacks, and rather than opposed, they could be combined such as done for Africa in \citet{Hernandez-Diaz2017}."

Finally, please go through the text again to check for language and consistency. E.g. Table 6 seems nowhere referenced in the text. You often confuse that and than. Etc. I hope that your co-authors can help with that too.

Authors' reply: Two co-authors went carefully trough the whole text again, checked for the languages, the consistency (use of projections vs scenarios, ...). We combined the sub-figures into single figures following recommendations from Copernicus, etc The number of changes made here is to large for them to be exhaustively listed here and we invite to refer to the marked up modification version of the manuscript.

Please provide replies to my points and mark your modifications in the manuscript. Thank you and best regards

Christian Haas

[revised manuscript text omitted]